behaviour, ecology

fin whale, song, food-associated call, prey biomass, ecosystem model

**Author for correspondence:**
Miriam Romagosa
e-mail: m.romagosa4@gmail.com

# Food talk: 40-Hz fin whale calls are associated with prey biomass

Miriam Romagosa[1], Sergi Pérez-Jorge[1], Irma Cascão[1], Helena Mouriño[2], Patrick Lehodey[3], Andreia Pereira[4], Tiago A. Marques[5,6], Luís Matias[4] and Mónica A. Silva[1]

[1]Okeanos – Instituto de Investigação em Ciências do Mar, Universidade dos Açores & IMAR – Instituto do Mar, Horta, Portugal
[2]Centro de Matemática, Aplicações Fundamentais e Investigação Operacional, Faculdade de Ciências, Universidade de Lisboa, Lisboa, Portugal
[3]Collecte Localisation Satellite (CLS), Ramonville St Agne, France
[4]Instituto Dom Luiz (IDL), Universidade de Lisboa, Lisboa, Portugal
[5]Centro de Estatística e Aplicações, Departamento de Biologia, Faculdade de Ciências, Universidade de Lisboa, Lisboa, Portugal
[6]Centre for Research into Ecological and Environmental Modelling, University of St Andrews, St Andrews, UK

MR, 0000-0003-2781-5528; SP-J, 0000-0002-4843-0443; IC, 0000-0001-6231-0483; HM, 0000-0001-7606-9643; PL, 0000-0002-2753-4796; AP, 0000-0002-5368-5707; TAM, 0000-0002-2581-1972; LM, 0000-0002-8086-4874; MAS, 0000-0002-2683-309X

Animals use varied acoustic signals that play critical roles in their lives. Understanding the function of these signals may inform about key life-history processes relevant for conservation. In the case of fin whales (*Balaenoptera physalus*), that produce different call types associated with different behaviours, several hypotheses have emerged regarding call function, but the topic still remains in its infancy. Here, we investigate the potential function of two fin whale vocalizations, the song-forming 20-Hz call and the 40-Hz call, by examining their production in relation to season, year and prey biomass. Our results showed that the production of 20-Hz calls was strongly influenced by season, with a clear peak during the breeding months, and secondarily by year, likely due to changes in whale abundance. These results support the reproductive function of the 20-Hz song used as an acoustic display. Conversely, season and year had no effect on variation in 40-Hz calling rates, but prey biomass did. This is the first study linking 40-Hz call activity to prey biomass, supporting the previously suggested food-associated function of this call. Understanding the functions of animal signals can help identifying functional habitats and predict the negative effects of human activities with important implications for conservation.

## 1. Introduction

Animals produce an array of different acoustic signals. These signals can encode various types of information about the signaller's attributes or external environment, and serve various purposes. During the mating season, males of many species produce high intensity and repetitive songs to attract or court females, repel conspecific males, or both [1–3]. It has been suggested that male songs can convey information about the individual's reproductive status, body size or health [4,5] and may be used by females and other males to assess the signaller's quality and competitiveness [6–8]. Numerous bird and mammal species produce food-associated calls. These calls can hold information on type, quality or quantity of food available and be used to announce resource ownership or attract others to the food source [9]. Many species give alarm calls in response to particular predators or predator abundance, thereby informing conspecifics about a threat [10,11], while social contact calls are often

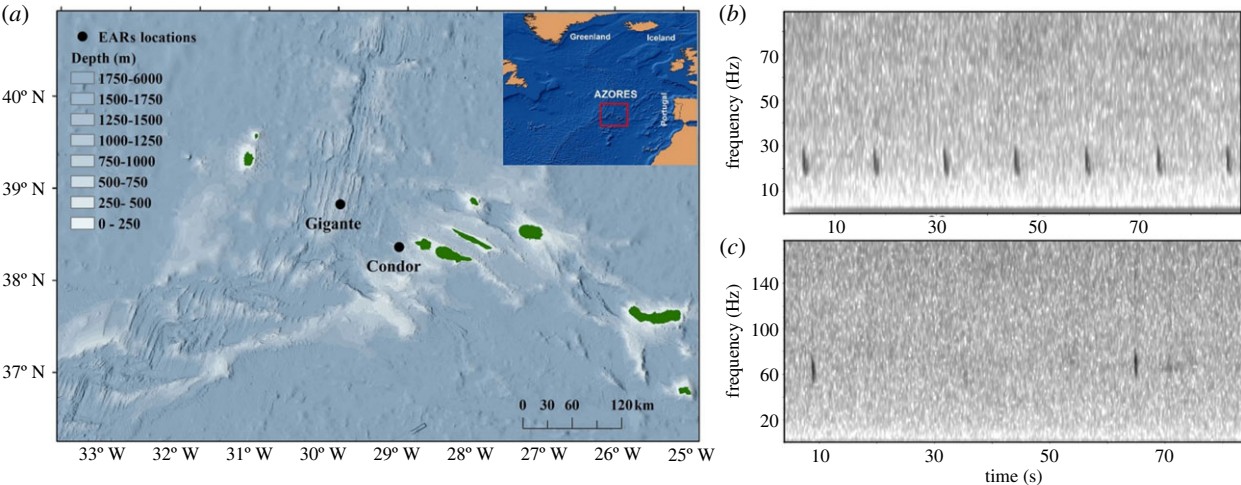

**Figure 1.** (*a*) Location of the Azores (inset map) and of the hydrophone moorings (black dots) at two locations (Gigante and Condor). Example spectrograms showing (*b*) the 20-Hz and (*c*) the 40-Hz call. (Online version in colour.)

used to maintain group cohesion, coordinate group activities and mediate social interactions [12,13]. As animal's acoustic signals play a critical role in their reproduction and survival, understanding the context of production and information content of these signals can give valuable insights into key life-history processes relevant for conservation [14].

Fin whales (*Balaenoptera physalus*) produce distinct vocalizations but knowledge about the functions of their calls is still limited. The most reported fin whale call worldwide is the 20-Hz note [15–19], a short-frequency downsweep mostly centred around 20 Hz [15]. The 20-Hz call can be produced (i) in regular sequences, forming a stereotypical song [15,16]; (ii) at irregular intervals [20] and (iii) as counter-calls [21]. Songs have only been documented from males [22] and are produced mainly during the known breeding season of the species [23,24]. Thus, it has been hypothesized that male fin whale song is used to attract females, either as an acoustic display [15] or by advertising patchy food resources [22]. Non-song counter-calling and irregular 20-Hz calls are normally produced by animals in groups [20] and probably serve a social function, such as maintaining contact with moving conspecifics [21,25]. Fin whales also produce a 40-Hz call sweeping in frequency from 75 Hz to 40 Hz [20,26,27]. The 40-Hz call is mostly detected in late spring and summer in known feeding areas [26], in association with complex topographical features [28,29] and feeding behaviours [20], suggesting a potential food-associated function.

To investigate the hypothesized fin whale call functions, we examine variation in production rates of song-forming 20-Hz calls and 40-Hz calls with respect to season, year and prey biomass. If males use 20-Hz calls to attract females through acoustic display [15], we expect call production to be mainly driven by season, peaking in winter, the known mating period of the species [23,24]. If, on the other hand, the 20-Hz call is used to attract females via food advertising [22], singing activity will be influenced by both season and prey biomass. Finally, if the 40-Hz call is associated with foraging activity, we predict that calling rates will be positively related to prey biomass, comparable with foraging calls of other species (e.g. bottlenose dolphin (*Tursiops truncatus*) bray calls, humpback whale (*Megaptera novaeangliae*) 'megapclicks') [30,31]. To test these predictions, we used a five-year acoustic dataset from bottom-moored hydrophones to extract call rates of each call type. In the absence of concurrent measurements of prey biomass, an ecosystem model was used to provide hindcast simulations of low trophic level (mesozooplankton) biomass for the area and period of acoustic recordings [32,33]. This approach allowed investigating the direct relationship between fin whale vocal behaviour and predicted prey biomass, avoiding interpretation of relationships with time-lagged prey proxies (i.e. chlorophyll).

## 2. Methods

### (a) Acoustic data collection and analyses

Passive acoustic monitoring (PAM) data were collected at two locations off the Azores Archipelago (figure 1*a*) using ecological acoustic recorders (EARs) [34] deployed at depths of approximately 200 m. The EAR consists of a Sensor Technology SQ26-01 hydrophone with a response sensitivity ranging from 193 to 194 dB re 1 V/μPa (depending on deployments) and a flat frequency response (±1.5 dB) from 18 Hz to 28 kHz. Hydrophones recorded from March 2008 to October 2012 with several data gaps and duty cycles (electronic supplementary material, figure S1). Despite gaps in the acoustic dataset, all seasons were well represented across the five sampled years.

Acoustic recordings were analysed for two fin whale vocalizations: the 20-Hz call, a 1 s downsweep centred at 20 Hz [15] (figure 1*b*) and the 40-Hz call, a 0.3 s downsweep from 75 Hz to 40 Hz occurring in irregular sequences [20] (figure 1*c*). All acoustic data were downsampled to 1 kHz to facilitate analysis. The 20-Hz call was previously analysed from these recordings and for another study [35] by using the low-frequency detection and classification system (LFDCS) [36]. Based on a reference call library of manually identified 20-Hz fin whale calls, the LFDCS detected candidate calls and estimated their pitch-track, which characterizes the frequency and amplitude variation of the signal over time. Each candidate call was compared to the reference library using a quadratic discriminant function analysis (QDFA). LFDCS performance was assessed by comparing detector outputs with manually analysed notes, yielding 0.9% of false positives, 80% of true positives and 20% of missed calls (for more details on the methodology [35]).

Three months (representative of each season) with longer duty cycle recordings (1 h of continuous recordings) were manually inspected to identify song and non-song 20-Hz calls. Results

showed that only 2.5% of the files contained non-song 20-Hz calls (Oct: 0%; Nov: 3.5% and Mar: 0%). Thus, we assumed that most 20-Hz calls analysed in this study were part of songs. Identification of 40-Hz calls using automatic detectors is challenging because of the frequency overlap with sei (*Balaenoptera borealis*) and blue whale (*Balaenoptera musculus*) calls [37]. So, 40-Hz calls were detected by visually inspecting spectrograms of the entire dataset (2048-point FFT, Hanning window with 50% overlap) using Adobe Audition 3.0 (Adobe Systems Inc., San Jose, CA) and annotating each call. The 40-Hz call was identified from its acoustic characteristics [20,26], which clearly differentiates it from the 20-Hz call because of the higher frequencies that downsweep from 75 Hz to 40 Hz over 0.3–1 s. The 40-Hz call was also easily distinguished manually from blue whale D calls, previously identified in this dataset [35], as having a distinctly broader bandwidth and longer duration.

A call rate index was calculated as the total number of 20-Hz or 40-Hz calls detected in a week divided by the recording time, in hours, during that week, to reduce potential bias from the different duty cycles. Hereafter, we will in general refer to 20-Hz or 40-Hz call rates, but these strictly mean the corresponding call rate index.

## (b) Zooplankton model

Stable isotope analysis of skin and faeces indicates that fin whales from the study area feed primarily on zooplankton (mainly euphausiids and copepods) [38,39]. In addition, meso-zooplankton biomass derived from a spatial ecosystem and population dynamics model (SEAPODYM) was the most important predictor of the distribution of fin whales in the Azores and across the mid-North Atlantic, while micronekton biomass estimates from the same model had no effect on the movements of the species [40]. Thus, we assumed that zooplankton is the main prey of fin whales in the study area and obtained estimates of zooplankton biomass from the lower trophic level SEAPODYM model (SEAPODYM-LTL) [32,33]. The SEAPODYM-LTL is a spatially explicit ecosystem and population dynamics model that simulates the biomass of mesozooplankton organisms within the epipelagic layer defined by the euphotic depth. The model is driven by physical and biological variables and applies a series of advection–diffusion–reaction equations [32]. Physical variables (temperature and currents) were extracted from the ocean reanalysis GLORYS (https://www.mercator-ocean.fr/en/ocean-science/glorys/), produced with the ocean general circulation model NEMO (http://www.nemo-ocean.eu/), in an eddy-permitting configuration [41–43]. Net primary production and euphotic depth were derived from ocean colour satellite data (http://www.science.oregonstate.edu/ocean). Outputs from these models were interpolated onto a weekly timescale and a spatial resolution of $0.25° \times 0.25°$ to be used by the SEAPODYM-LTL model. The model predicts weekly mesozooplankton biomass on a global spatial grid ($0.25° \times 0.25°$). Predictions for the period 1998–2019 are publicly available (https://marine.copernicus.eu/). The model validation is based on the climatological database COPEPOD that provides standardized mean zooplankton biomass values on a global spatial grid [42,43].

## (c) Spatial scale of data integration

SEAPODYM-LTL estimates of mesozooplankton biomass (hereafter zooplankton biomass) were extracted for the weeks with acoustic recordings (electronic supplementary material, figure S1) and averaged across $0.25° \times 0.25°$ grid cells centred around the hydrophone position. To determine the most appropriate spatial scale (i.e. the number of grid cells) for analysing SEAPODYM-LTL data in relation to acoustic data, the maximum detection range of 20-Hz and 40-Hz fin whale calls was estimated using the sonar equation [44]:

$$SNR = SL - TL - NL + 10 \log_{10} BW,$$

where SL is the transmitted source level (dB rms re 1 µPa at 1 m), TL is one-way transmission loss (dB), NL is the ambient noise level at the receiver (dB rms re 1 µPa) and BW is the processing bandwidth (Hz). Source levels of 20-Hz and 40-Hz calls were calculated using calls localized by three EARs deployed in a nearby area. The propagation range-dependent acoustic model (RAM) [45,46] was used for the calculation of TL. Ambient NL were calculated for the frequency band of each call type and for the quietest and noisiest months within the recording period (see electronic supplementary material, text S1B for more details).

Finally, a sensitivity test of the scale of data integration was performed by analysing annual and monthly patterns of estimates of zooplankton biomass at a range of scales.

## (d) Statistical analyses

Data from summer months (June to August) were excluded because the summer matches the end of fin whale migration through the Azores and whales are rare in the area [40,47,48]; hence the lack of acoustic detections in the summer simply reflects the absence of whales and not changes in calling patterns.

The quasi-Poisson model (a particular case of a generalized linear model, GLM) was used to describe the relationship between 20-Hz and 40-Hz call rates, and a set of independent variables: year, season (according to meteorological definition) and zooplankton biomass. This modelling tool is especially suited to handle overdispersed count variables because it incorporates an overdispersion parameter that allows for more spread than the standard Poisson mean–variance relationship [49,50]. Separate models were built for the 20-Hz and 40-Hz call rates to understand how the same explanatory variables affected each vocalization type. The variance inflation factor (VIF) was calculated for the complete models to measure the strength of correlation between all predictor variables (season, year and zooplankton biomass). VIF values higher than 5 or 10 are considered too high and could cause misinterpretation of model outputs [51]. In our models, VIF values for the three variables were approximately 1. Season and year were used to assess intra and inter-annual variations in the response variables. Given that zooplankton biomass varied with season, an interaction between these two variables was also included in the models. No interaction between season and year was included because seasonality in calling did not vary with year. Detailed descriptions of the statistical models are given in the electronic supplementary material, text S1C.

The best model was selected based on the lowest quasi-Akaike's information criterion (QAIC). Drop-in deviance tests (based on the *F*-test to account for overdispersion) were computed for the best model to validate each explanatory variable's statistical significance. The process started with the null model, and each explanatory variable was added sequentially until reaching the best model defined by the QAIC criterion. Moreover, the Wald test was applied to each parameter of the model to test the null hypothesis that the respective parameter is equal to zero.

Model assumptions were verified by plotting residuals versus fitted values to check for heterogeneity of variance and residual QQ plots to check for normality. Half-normal probability plots of the residuals with simulated envelope were computed [52] to check whether the choice of the random component of the model was appropriate and identify possible outliers in the data [53]. Additionally, the temporal dependency of the residuals was assessed [54] to detect autocorrelation in

Proc. R. Soc. B **288**: 20211156

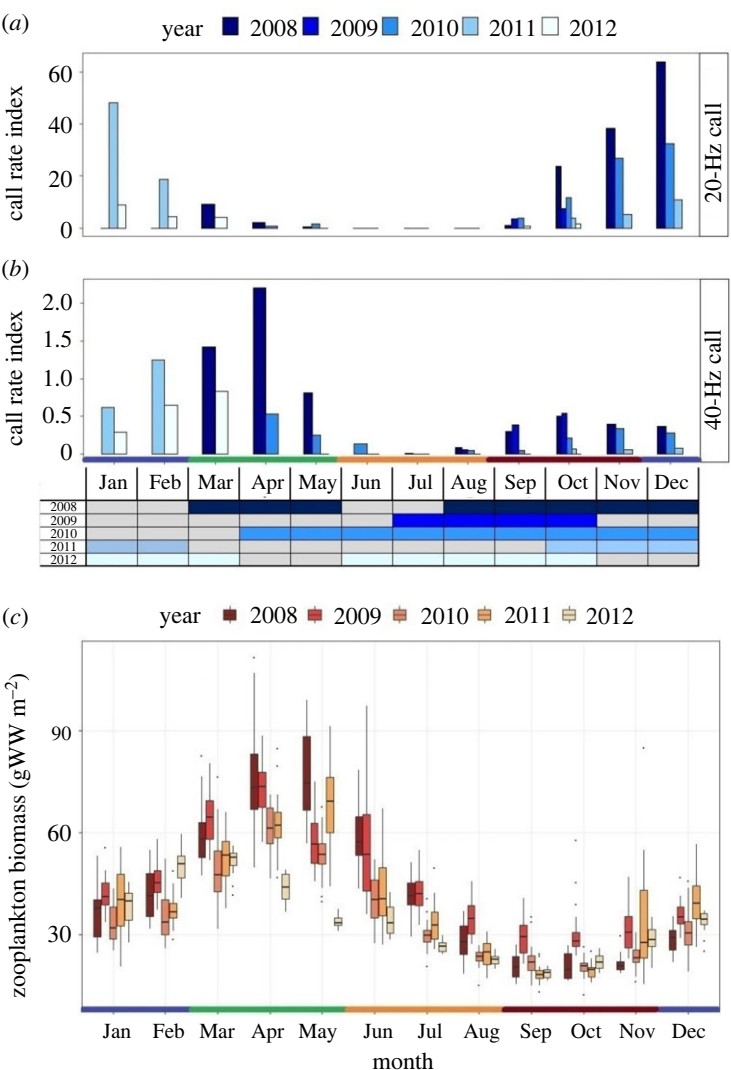

**Figure 2.** Monthly variation in (*a*) 20-Hz and (*b*) 40-Hz call rates, and (*c*) model-based estimates of zooplankton biomass (gWW: grams wet weight), from 2008 to 2012. The graph on the *x*-axis in (*a*) and (*b*) represents the recording effort by month and year and grey colour indicates no data. Horizontal lines within the boxplots in (*c*) indicate the median, box boundaries indicate the 25th (lower boundary) and 75th (upper boundary) percentiles, vertical lines indicate the largest (upper whisker) and smallest (lower whisker) values no further than 1.5 times the interquartile range, and black dots represent outliers. Colours on the *x*-axis indicate seasons: blue, winter; green, spring; orange, summer; brown, autumn. Abbreviations for months are the following: Jan, January; Feb, February; Mar, March; Apr, April; Jun, June; Jul, July; Aug, August; Sep, September; Oct, October; Nov, November; Dec, December. (Online version in colour.)

the data. An autocorrelation at lag 1 was detected for the 20-Hz call rates, implying there was a correlation between call rates in successive weeks. To account for the temporal autocorrelation, one-week lagged values of 20-Hz call rates were included in the model as a predictor variable. All statistical analyses were performed using the software R (v. 4.0.2) [55].

## 3. Results

### (a) Detection range and zooplankton biomass spatial scale

Median detection ranges at the deployment locations were 64 km for the 20-Hz call and 18 km for the 40-Hz call (electronic supplementary material, table S1). Therefore, SEAPODYM-LTL estimates of zooplankton biomass extracted for the weeks with acoustic recordings (electronic supplementary material, figure S1) were averaged across four grid cells of 0.25° × 0.25° centred around the hydrophone position (55 × 55 km). Changing the number of grid cells to nine (83 × 83 km) or 16 (194 × 194 km) had little or no effect on the annual and monthly patterns of

estimated zooplankton biomass (electronic supplementary material, figure S2).

### (b) Temporal occurrence of calls and zooplankton biomass

Rates of the 20-Hz call increased in autumn, peaked in winter, decreased in spring and were null in summer (figure 2*a*). Conversely, 40-Hz call rates were low in autumn, increased in late winter, reached highest values in spring and decreased again in summer (figure 2*b*). Zooplankton biomass showed a clear peak in spring (April–May), decreased throughout the summer and early autumn and increased again in winter (figure 2*c*).

### (c) Model of the 20-Hz call

Season was the most important predictor of the 20-Hz call, followed by year and one-week lagged call rates (57% deviance explained; electronic supplementary material, table S2). Zooplankton biomass had no significant effect on

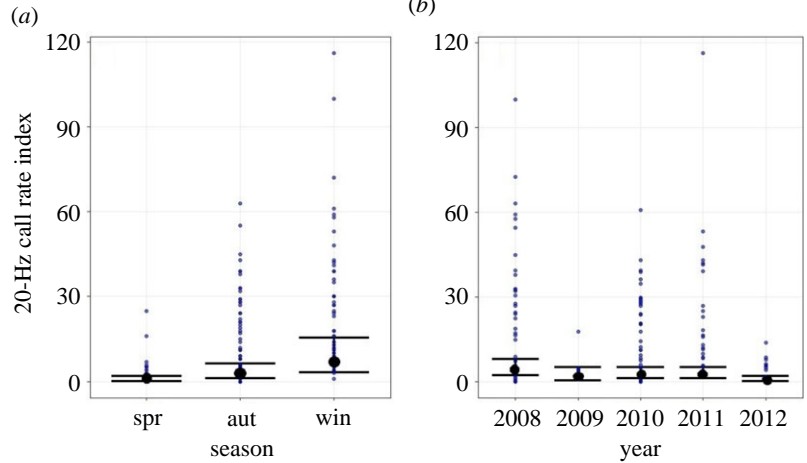

**Figure 3.** Results from the quasi-Poisson model showing the (*a*) season and (*b*) year effect on the 20-Hz call rate. Blue points represent observations, error bars represent the mean (back dot) and 95% confidence intervals of fitted values. Seasons are abbreviated as follows: spr, spring; aut, autumn; win, winter. (Online version in colour.)

**Table 1.** Analysis of deviance (ANOVA) for the best fitting quasi-Poisson model of the 20-Hz call rate. Significant terms ($p < 0.05$) are shown in italics. d.f.—degrees of freedom, *Dev.*—deviance, *Res.* d.f.—residual d.f., *Res dev.*—residual deviance.

| | d.f. | *Dev.* | *Res.* d.f. | *Res dev* | *F* | *p*-value |
|---|---|---|---|---|---|---|
| NULL | | | 142 | 3066.7 | | |
| season | 2 | 955.38 | 140 | 2111.3 | 47.25 | *<0.001* |
| year | 4 | 601.36 | 136 | 1510.0 | 14.87 | *<0.001* |
| lag-1-call rate | 1 | 209.92 | 135 | 1300.0 | 20.76 | *<0.001* |

**Table 2.** Analysis of deviance (ANOVA) for the best fitting quasi-Poisson model of the 40-Hz call rate. Significant terms ($p < 0.05$) are shown in italics. d.f.—degrees of freedom, *Dev.*—deviance, *Res.* d.f.—residual d.f., *Res dev.*—residual deviance.

| | d.f. | *Dev.* | *Res.* d.f. | *Res dev* | *F* | *p*-value |
|---|---|---|---|---|---|---|
| NULL | | | 143 | 197.81 | | |
| zooplankton | 1 | 39.62 | 142 | 158.18 | 25.94 | *<0.001* |

the 20-Hz call (table 1; electronic supplementary material, table S3). The call rate was significantly higher in winter than in autumn and spring but did not differ between these later seasons (figure 3*a*). 2012 had significantly lower call rates than all other years except 2009 (figure 3*b*; electronic supplementary material, table S3). Overall, the model residuals did not show any pattern, indicating a good fit to the data. Most of the residuals were within the simulated envelope (electronic supplementary material, figure S3).

### (d) Model of the 40-Hz call
The best model for the 40-Hz call rate included only zooplankton biomass (20% deviance explained; electronic supplementary material, table S2). Call rate increased with increasing zooplankton biomass (table 2 and figure 4; electronic supplementary material, table S3). Although zooplankton biomass varied seasonally (figure 2*c*), the interaction between these two variables had no significant effect on 40-Hz call rates. Model residuals did not show outliers and indicated the model was adequate to describe the data (electronic supplementary material, figure S4).

## 4. Discussion
Our study shows that the production of 40-Hz calls in fin whales is positively associated with prey biomass, providing supporting evidence of a food-associated signal, as previously suggested [20,26–29]. The 40-Hz call rates increased with increasing biomass of zooplankton, the main component of the fin whale diet [38,39]. Conversely, the production of 20-Hz calls was mainly influenced by season and to a lesser extent by year, but temporal patterns were independent of zooplankton biomass. This finding corroborates the widely accepted view that 20-Hz songs are used in a reproductive context [15,22] but suggests their function is independent of food biomass.

### (a) 20-Hz song function
The winter peak in 20-Hz calls found in this study is consistent with the known seasonality of the fin whale song in the Northern Hemisphere [15,16,35,56]. The fact that the 20-Hz song peaks during the breeding season of the species [23,24], is produced only by males [22] and is well-suited for long-range communication [57], support the widely

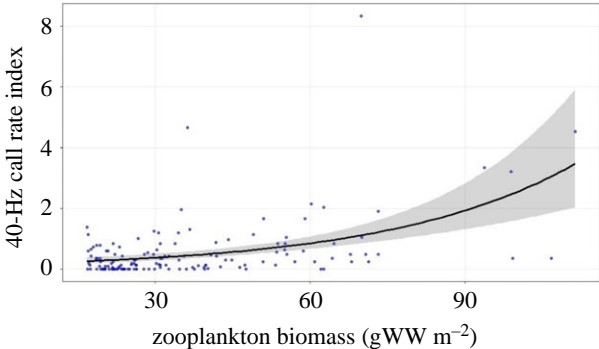

**Figure 4.** Zooplankton biomass effect on the 40-Hz call. Blue points represent the observed 40-Hz call rate index. The solid line corresponds to the mean fitted values (i.e. the trend) and the grey shaded area represents the 95% confidence interval. (Online version in colour.)

accepted hypothesis that fin whale song is a male advertisement signal [15,22]. Similarly, it has been suggested that male fin whales sing to attract females, mediate interactions with other males, or a combination of both [15,22,58]. Croll *et al.* [22] proposed that male fin whale song could attract females by conveying information on aggregations of patchily distributed prey. In the resident fin whale population of the Gulf of California [59], male singing co-occurs with winter foraging on dense aggregations of krill [22,60]. In this study, zooplankton biomass had no effect on fin whale singing activity, as would be expected if male songs signal food aggregations. Similarly, other studies did not find a relationship between prey proxies (i.e. acoustic backscatter strength) and the production of song-forming 20-Hz calls [29,61]. Thus, findings from this and previous studies do not support the hypothesis that fin whale song serves to attract females via food advertising. Instead, these results are in line with the hypothesis that singing may be an acoustic display [15] directed towards females or other males. Evidence from a range of taxa indicates that songs can convey honest information on singer's motivation and quality which may be used both by females in mating decisions, and by other males in competitive interactions [4,5]. However, there are also species in which song traits do not correlate with male quality [62] and further research is needed to directly test this hypothesis in fin whales.

The effect of year on singing activity was greatly influenced by 2012, which showed significantly lower call rates when compared to 2008, 2010 and 2011. The most plausible explanation is that decreased call rates in 2012 reflect lower numbers of fin whales in the area. This is partly supported by visual observations collected by the Fisheries Observer Program Data showing that fin whale encounter rates (sightings/100 km) were null in 2012, compared to 0.9 in 2010 and 9.2 in 2011. Year-to-year variability in fin whale numbers is usually attributed to environmental changes affecting prey distribution and abundance [37,63]. In this study, models did not show an effect of prey biomass on singing activity. In addition, lower values of modelled zooplankton biomass in 2012 were only found in April and May (figure 2c), the end of the singing season. It could also be argued that inter-annual variability in call rates results from differences in call detectability due to variations in background noise from shipping. Although noise levels in the study area did not vary significantly between 2010 and 2012 [64], effects of shipping noise on call detectability should be investigated in the future.

## (b) 40-Hz call function

This study confirms the temporal separation between fin whale 40-Hz calls and 20-Hz calls [26]. More importantly, we demonstrate that production of fin whale 40-Hz calls was best predicted by zooplankton biomass alone across all years and seasons, with call rates increasing with increasing prey biomass. These results lend support to previous suggestions of a food-related function of the 40-Hz call [20,26,28,29]. In the eastern North Pacific, 40-Hz calls peaked in early summer at known important feeding habitats [26]. In the Canadian Pacific, distance from the shelf break and backscatter intensity (as the proxy of potential prey) were important determinants of fin whale 40-Hz calls [29]. In addition, 40-Hz calls were generally produced by whales in groups, engaged in long, possibly foraging, dives [60] or surface feeding activities [20]. Together, findings from this and previous studies provide strong evidence for the use of 40-Hz calls in a feeding context.

Some food-associated calls are produced only in feeding contexts, with animals adjusting call types or rates as a function of the type, quality or quantity of food available [65]. More commonly, food-associated calls are given in multiple contexts and are not food-specific [9]. Irrespective of their degree of context-specificity, there is increasing evidence that food-associated calls provide receivers with information about a food source or feeding event and often are used to attract them to a foraging site. In many cases, food-associated calling functions to recruit potential mates or kin, increasing the inclusive fitness of callers [66], or to recruit non-related partners and allies, potentially enhancing social status and bonds [67]. Attracting conspecifics to a feeding site may also increase the foraging efficiency of callers, by facilitating prey capture or defence, or helping with predator vigilance [68]. There are also examples where food-associated calls are not used to attract others but to reduce or mediate competitive interactions over food by establishing resource ownership [69]. Clearly, the ultimate function of food-associated calling varies greatly with the social and ecological environment of animals [9].

Fin whales do not live in stable social groups [70] and the distribution of their prey is ephemeral [71]. Thus, it is unlikely that fin whale 40-Hz calls serve to attract kin or social partners, either to provide them with increased foraging benefits or to assist defending food patches. Also, the lower detections of fin whale 40-Hz calls during the breeding season reported here and in other studies [20,26], suggests that the primary function of this call is not to attract potential mates, trading-off food for reproductive benefits. In other cetaceans, food-associated vocalizations have been recorded during cooperative foraging behaviours (e.g. humpback whales [72], killer whales (*Orcinus orca*) [73]) and may assist with prey herding and capture [74]. With the exception of a single report of fin whales feeding at the surface in perfect synchrony [75], there is no evidence of cooperative feeding in fin whales. Nevertheless, attracting other whales to the foraging site may increase the chances of tracking prey movements, thus prolonging feeding opportunities for callers, as suggested for cliff swallows (*Petrochelidon pyrrhonota*) feeding on insect swarms [76]. Fin whales often occur in temporary

foraging aggregations in our study area and elsewhere [26,47]. Fin whale 40-Hz calls could be used to convey information about the individual location to regulate spacing between foragers, or establish ownership of food patches, as described for other species [69]. At present though, we do not know the functional significance(s) of the 40-Hz fin whale call when produced in feeding contexts.

The recent description of two acoustically tracked fin whales producing 40-Hz calls while moving past each other [27], gives some indication that this call might also serve as a contact or social call. In birds and mammals, functionally specific vocalizations, like food or alarm calls, are often used in different behavioural contexts [9]. Blue whale 'D calls' were firstly described as food-associated and social calls because, as the 40-Hz fin whale calls, were recorded during foraging behaviours in feeding areas [77,78] and from whales in groups [79,80]. Later though, one study reported D calls also produced in a reproductive context where two males were aggressively interacting with each other while escorting a female [81]. Thus, it is likely that more functions for the 40-Hz call may be revealed with the increasing research effort on fin whale vocal behaviour.

## 5. Conclusion

Our study is the first to show a positive association between the production of the 40-Hz call and modelled biomass of prey, providing additional evidence of the use of this call in feeding contexts. Our findings are also consistent with earlier work indicating that the song-forming 20-Hz call is used in reproductive contexts, but the absence of a relationship with prey biomass does not support the assumption that this call is used by males to advertise a food source and attract potential mates. Instead, the 20-Hz song may be a male acoustic display used in intersexual and intrasexual interactions. Our study also illustrates how spatio-temporally resolved simulations of zooplankton biomass, which is challenging to measure in the field, can provide valuable insights into the environmental context and potential functions of baleen whale vocalizations.

Understanding call function and monitoring vocal behaviours associated with the state of individuals or groups (e.g. reproductive status and success, and social complexity), habitat quality (e.g. food resources) or animal density (e.g. call rates) can help identify functional habitats, predict negative human impacts and support conservation planning [14]. Information on the temporal and spatial occurrence of fin whale 40-Hz calls may inform when and where animals engage in foraging and provide important clues to the environmental factors promoting foraging behaviour on this

species. Similarly, the 20-Hz song may give unique insights into the location and characteristics of the areas used for mating. Studies combining visual and acoustic observations of callers and receivers simultaneously, offering information on the behavioural context of call production along with responses of conspecifics, could significantly advance our understanding of fin whale vocal behaviour.

Data accessibility. Datasets used in the analyses of this paper are available in the electronic supplementary material and from the Dryad Digital Repository: https://doi.org/10.5061/dryad. 00000003s [82]. Mesozooplankton model outputs (netcdf format) are available on the Copernicus Marine Service (https://marine. copernicus.eu/; GLOBAL_REANALYSIS_BIO_001_033) and are updated twice a year (next 2021 release will have resolution 1/12° x day).

Authors' contributions. M.R.: conceptualization, data curation, formal analysis, methodology, writing-original draft, writing-review and editing; S.P.J.: data curation, formal analysis, methodology, writing-review and editing; I.C.: investigation, methodology, writing-review and editing; H.M.: formal analysis, methodology, writing-review and editing; P.L.: methodology, resources, writing-review and editing; A.P.: formal analysis, methodology, writing-review and editing; T.A.M.: conceptualization, methodology, supervision, writing-review and editing; L.M.: formal analysis and methodology; M.A.S.: conceptualization, funding acquisition, investigation, project administration, supervision, writing-review and editing.

All authors gave final approval for publication and agreed to be held accountable for the work performed therein.

Competing interests. We declare we have no competing interests.

Funding. This work was supported by Fundação para a Ciência e Tecnologia (FCT) Azores 2020 Operational Programme and Fundo Regional da Ciência e Tecnologia (FRCT) through research projects TRACE (PTDC/ MAR/74071/2006), MAPCET (M2.1.2/F/012/ 2011) and AWARENESS (PTDC/BIA-BMA/30514/2017), co-funded by FEDER, COMPETE, QREN, POPH, ERDF, ESF, the Lisbon Regional Operational Programme and the Portuguese Ministry for Science and Education. Okeanos R&D Centre is supported by FCT through the strategic fund (UIDB/05634/2020). M.R. was supported by a DRCT doctoral grant (M3.1.a/F/028/2015). S.P.J. was funded by EC funds (SUMMER H2020-EU.3.2.3.1), I.C. by FCT through AWARENESS – (PTDC/BIA-BMA/30514/2017). H.M. acknowledges support by CMAF-CIO (funded by FCT, Portugal, through the projects UID/MAT/00006/2013 and UIDB/04561/2020, respectively). A.P. was supported by AWARENESS project (PTDC/BIA-BMA/30514/ 2017) and UIDB/50019/2020 – I.D.L. and T.A.M. by CEAUL and the LMR ACCURATE project (contract no. N3943019C2176). M.A.S. was funded by FCT and EC funds (IF/00943/2013, SUMMER H2020-EU.3.2.3.1, GA 817806).

Acknowledgements. We are grateful to Marc Lammers, for providing the EARs and technical support, and to Rui Prieto, Sérgio Gomes, Norberto Serpa, skilled skippers and crew that participated in the preparation and deployment of the EARs at DOP/IMAR. We also thank the reviewers that significantly improved the original manuscript. Data on fin whale sightings from 2010–2012 were kindly provided by the Azores Fisheries Observer Program (POPA) (www.popaobserver.org).

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
