## [Peer Review File · Proceedings of the Royal Society B: Biological Sciences]

Review History

RSPB-2021-0288.R0 (Original submission)

Review form: Reviewer 1

Recommendation

Major revision is needed (please make suggestions in comments)

Scientific importance: Is the manuscript an original and important contribution to its field?

Acceptable

General interest: Is the paper of sufficient general interest?

Acceptable

Quality of the paper: Is the overall quality of the paper suitable?

Marginal

Is the length of the paper justified?

No

Should the paper be seen by a specialist statistical reviewer?

No

Do you have any concerns about statistical analyses in this paper? If so, please specify them explicitly in your report.

Yes

It is a condition of publication that authors make their supporting data, code and materials available - either as supplementary material or hosted in an external repository. Please rate, if applicable, the supporting data on the following criteria.

Is it accessible?

Yes

Is it clear?

No

Is it adequate?

No

Do you have any ethical concerns with this paper?

No

Comments to the Author

General

Some rewriting for clarity and conciseness is needed throughout. There are also mis-matched tenses and singular/plural nouns in places. Several long, run on sentences

Much of the paper was overly wordy and presented unneeded details in some places, and missed critical detail in others. In some places wording was awkward. Read through and edit for conciseness and clarity.

Some pertinent details in the data acquisition and analysis are missing

This is modelled prey data tied to automated detected calls. How the data was entered into the model (its format) was missing, and other variables could have been included or current variables refined. All the variables have the possibility of being autocorrelated and so - while I believe the conclusion - it is hard to see how the data presented here is a strong supporter or indicator of the suggested results.

Seems like the authors went into the analysis knowing the answer they expected/wanted - to confirm a singing season and then forage- based calling

A wider literature review is needed - while the authors draw on other species in their introduction, justification of methods and conclusions (which in some cases is ok) they are missing some literature from previous fin whale studies that would inform this study.

Manuscript is long for page limit - and there is a lot of unnecessary material presented
Supplementary materials not mentioned in text

Abstract

Line 23 - perhaps: Vocalizations in animals show a great variety, with a variety of forms and suggested functions

Line 23- and shows

Line 26 - and whose function - does not read smoothly

Line 29 - mixed tenses

Line 31 - reproductive function

Line 30-32 - Did zooplankton biomass not vary with the season.

Line 33 - Be careful in saying this is the first - there are a number of studies that have suggested the foraging application of the 40-Hz call, and some linking zooplankton measures or proxies to calling. This statement is made throughout.

Line 35 - what do you mean by 'content' here. Do you mean structure, or use perhaps

Line 36 – use and not uses

Introduction

Line 44 – reads awkwardly

Line 47 – calls rather than sounds perhaps. I find the authors definition of calls and songs is odd. It may be just purely word choice, but do calls not form song?

Line 48 – are all reproductive displays from males when vocal described as song?

Line 48-49 – these statements need references

Line 50-51 – perhaps some more examples here would help, this is a very simplified and arbitrary distinction of animals that produce song

Line 53 – are songs not comprised by calls? Again the distinction of calls and song is not one I recognise or would support.

Line 56-57, and Line 61 – consider rephrasing

Line 62 – double spacing between ‘individuals.’ and ‘Yet’

Line 65-66 – reads awkwardly

Line 75 – authors say a variety of calls, and then describe just 2, one of which they have previously called a sound.

Line 81 – I believe the references are misused here. Males were most often traveling when calling in reference 21.

Line 82 – counter-calling perhaps rather than call-counter

Line 83 – serves

Line 84 – calls sweep down from 100 Hz to 30 Hz. It is written here in reverse.

Line 88 – there are many studies on the potential function of calls of fin whales not referenced by the authors

Line 89-90 – reads awkwardly.

Line 94 – reference to mating period, but no description of how this is known, or really when this is

Line 99 – double space between megapclicks and).

Methods

As written this is not replicable to this or another area. Some of the justification of model use should have been presented in the introduction

Line 114-115 – gaps in the data are mentioned but not presented, and the reader is simply told it will not have an impact on the analyses

Line 120 – song patterning could still be possible for short periods – and determined if not through the detector through manual analysis. Indeed, Figure 1B clearly shows an example of regularly spaced 20-Hz calls labeled as song and the clip is less than 90 seconds in length.

Line 121 – more description of the methods should be presented here than purely referencing another paper.

122-125 – although it is likely a safe assumption it is an assumption non-the-less, without any verification stated to support it. This section should be this should be better described and clarified. Manual analysis likely help to determine song. Non-song calling has not really been described so far, and is not throughout the paper.

Line 29 – why was manual analysis not done (or at least described in the methods) for 20 Hz calls

122-123 – a small proportion of the automated detected calls

Line 125 – when were the longer duty cycles in comparison to the shorter?

121-126 – was there no manual analysis on the 20-Hz calls – more details of the manual analysis is needed. For example was all data analysed or subsampled? Was this systematically done, guided by the detector

Line 120-126 – did the LFDCS not pick out the presence of 40-Hz calls

Line 135 – be consistent in the way you reference calls (20Hz, 40Hz are written 20-Hz and 40-Hz throughout the rest of the paper).

Line 142 – would be nice to see if/how the model was validated in its expression of zooplankton presence and abundance. Here there is no detail on the studied used to form this model –

whether it is spatially or temporally specific, and how many whales/seasons this sampling was completed over. For example, if there are no zooplankton would there be no whales or would they opportunistically eat fish?

Line 151-152 – could some of these physical variables be included rather than 3 variables that essentially represent the same thing.

Line 157-159 – could replace some of the details in 147-148. Rephrase so as a justification of model selection rather than sounding like a result

Line 142 – 159 – much more of this section needs description and justification for the reader

Line 162 – what allows certainty that the whale you are observing is the whale that is vocalising. Were any detection range calculations done to estimate call detection in different ambient noise settings. Detection range is very site dependent - there should be more detail here about how range was estimated/derived and what this range means.

Line 170-171 - I'm not sure, without whale survey data, how this differs from any other time when calls are not heard

Line 171 – is the lack of whales and lack of vocalisations seasonally, if not prey, driven?

Line 174 – you assess collinearity, but establish that there are not patterns in prey driven by season and/or year?

Line 174 – 175 – separate models were built for each call type using the call rate index. Also- why were separate models used?

177 – were these the only 3 variables in the mode? Why would calling differ by year or season alone? How was season defined

178 – details of the interaction variable included is needed – both here and in the results. I assume that zooplankton abundance and season are linked (see also in Figure 2) but at the moment it is unclear how the model will signal that the season is important and not prey or vice versa.

Reference supplementary materials.

185 – AIC – needs to be explained further, it is used in acronym twice. Perhaps these values should also be displayed

191 – this needs a plain language explanation to set in context and better explain to the reader what steps were taken and why

Line 194 – why was a lag of one week used. Do the authors believe that this difference in time would allow any autocorrelation in variables to be rectified, or is this based on a biological phenomenon, or example?

Results

Line 197-198 – consider rewording

Line 197 – is there any control for the number of whales that might be present singing in your rate of calling? There was no whale surveying described in the methods, or anything other than statements of when they were present in the study area or not.

Line 199-201 – I think it should be made clear that this is modelled biomass of prey

Line 202 – lower calling rate

Line 202 – Is this lower than all years – what was the inter-annual variability like? Was it significant between years? The authors have stated that said the interannual seasonal variation was negligible but the annual variation was not?

Figure 2A – 2012 showed the least calling but it and 2011 were the only years to show calling in Jan, Feb. There appears to be patterning between years in call presence/month that is not commented on

Figure 2C is the assumption that the zooplankton biomass is relatively stable in its patterned presence each year?

Figure 2C – the x axis needs a label

Figure 3 – the black dots are not clear, nor is what they represent

Figure 4 – most calls fall outside of the 95% percent confidence area.

Is Figure 2B suggesting that there were not even 2 calls/week in all but one case for 40-Hz calls?

Discussion

Line 241 – again, this is not the first study to have suggested or demonstrated this link or the distinction between call types

Line 243 – could seasonality not be highlighted as a factor of importance because at some periods you have song and others just 20 Hz calls. Perhaps some manual analysis to examine the patterning between calls rather than just number might help this

Line 249-250 – reads awkwardly

Line 256-258 – speculative here. Perhaps wording can make it less so, but the function of the call and the meaning of the calls are 2 very different things

Calling can be more limited when whales are focused on foraging – so type of call may reflect the prey presence, but number or rate may not

The pattern is not consistently seen between years – what was different between, say, 2008 and 2010 – and the other years

- were there any differences between years in the physical/oceanographic variables? What is underlying the differences in prey abundance annually?

2012 – no recovery in the latter season when the zooplankton abundance starts to recover

Line 257 – breeding and feeding occur together? Again is there evidence to support this for your area and times of study

Line 264 – there is the suggestion for humpbacks that calling while feeding is to advertise prey presence and become a more protracted form of courtship – but it doesn't seem to be what you are suggesting here

Line 262-274 – much of this doesn't seem relevant and isn't connected to what has been presented through the paper

Line 275 – 276 – not interannual prey variation? Here this argues too that year and prey are not independent variables.

Line 275 – annual variation mechanism should be described, as should how you know that whale numbers fluctuate

Line 280 – authors say may have affects – did it?

Line 291-292 – this is written as if observations were part of this study, but then reference other studies – rephrase.

307 – the call number seems low even when it is the dominant call source – perhaps more could be detailed in the results about call numbers in this study.

Line 314 – 316 – reads very awkwardly, like there is a word missing

Line 316 – 318 – the authors state what could be possible, but it seems like they had some of this information to hand and did not use it.

Line 325 – if you take into account the detection range.

Line 326-327 – what does different and appropriate mean here.

328 – 338 – not relevant to what is presented here – speculating on aspects of the work that is not examined. Song characteristic were not described at all in this work.

Conclusions – strongly worded and unfounded. Greater literature review needed from the authors before making this statement.

Review form: Reviewer 2

Recommendation

Accept with minor revision (please list in comments)

Scientific importance: Is the manuscript an original and important contribution to its field?

Good

General interest: Is the paper of sufficient general interest?

Acceptable

Quality of the paper: Is the overall quality of the paper suitable?

Acceptable

Is the length of the paper justified?

Yes

Should the paper be seen by a specialist statistical reviewer?

No

Do you have any concerns about statistical analyses in this paper? If so, please specify them explicitly in your report.

Yes

It is a condition of publication that authors make their supporting data, code and materials available - either as supplementary material or hosted in an external repository. Please rate, if applicable, the supporting data on the following criteria.

Is it accessible?

Yes

Is it clear?

Yes

Is it adequate?

Yes

Do you have any ethical concerns with this paper?

No

Comments to the Author

Note that manuscript central makes a mess of text formatting, so I have submitted this as a pdf as well.

This paper makes a real contribution in relating prey abundance to fin whale calling and it deserves to be published. However, there are some problems that require revision. I have corrected textual errors in the annotated pdf and many are listed in the detailed line-by-line comments, but I highlight some here. The text fails to mention some of the most relevant papers on seasonality of 20 and 40 Hz fin calls in the relevant places and needs to include them. The argument on lines 160-161 that fin calls are only detectable at ranges <50km is directly contradicted by the paper cited in the discussion on long range communication. Much better to include both data on Received Levels at different ranges with data on the Source Level of finback calls and transmission loss in your environment and depth of your recorders with respect to sound speed profile to estimate range of detection. It is essential for you to do a better job estimating range of detection and discussing consequences for the scale at which you analyse the data.

I am struck in Figure 4 that the relationship between call rate seems to depend highly on 3-4 points out of a large sample. Any specific info on the context for these? Perhaps worth discussing robustness of result if it is so dependent on a few outliers. I'd be interested to see a scatterplot of call rate by year and season included in the supplementary materials, especially since QAIC for models including these variables was so close to zool only.

Lines 272-314 contain some errors that I flag in the line-by-line

Once these problems addressed in revision, this paper should be ready for publication

Line-by-line comments

line 1 "prey availability" should be changed to "prey abundance" Measuring how available prey is for the whales would require much finer resolution of prey patches

Line 26 The abstract suggests only indirect evidence on functions of fin whale calls was available in past work and that this paper provides direct evidence. This is arguably true for the 40 Hz calls given link to prey here, but the evidence in support of reproductive function of 20 Hz series is no different from that of many past papers and paper should give clearer credit in intro to earlier papers linking 40 Hz calls to foraging season, e.g. Širović et al. 2012. The first line of the conclusion is more accurate.

Line 45: Duration and rhythm do not affect how far a sound can travel nor how well it reaches the receiver (not clear what the difference is between these 2 – are you referring to multi-path?). Longer duration and repeated predictable rhythm may improve ability to detect and classify the call in noise, but that is a different topic.

P49 I do not think most males sing to resolve conflict – it plays a role in male-male competition.
L 53: Females produce song in many species

L 56: what is evidence no calls convey info about the caller as opposed to environment? For example, a threat call may refer to the state of the caller not the environment.

L 65-66: I would argue that you cannot understand the function of a call without also understanding when and how the receiver responds.

L68: The context in which humpbacks sing and how others respond has been studied directly. Not the same for other whales, so does not make sense to lump them. This has been reported for 40Hz calling fin whales, e.g. Wiggins and Hildebrand 2020, so important to include in intro.

L 106 how is depth a proxy for prey and why does it require a time lag?

L 160-161 I think that your justification for detection range of finback calls is very weak, based mainly on unpublished survey data that cannot define detection ranges. detection ranges depend on transmission loss in habitat and depth of recorder. Many sites are predicted to have >200km detection range and in discussion you cite 49 which argues for many hundreds of km range. The short detection range for humpback song is not relevant for finback case. Much better to use data on the SL of finback calls and transmission loss in your environment and depth of your recorders with respect to sound speed profile.

Figure 4. The relationship between call rate seems to depend highly on 3-4 points out of a large sample. Any specific info on the context for these? Perhaps worth discussing robustness of result if it is so dependent on a few outliers. I'd be interested to see scatterplot of call rate by year and season, especially since QAIC was so close to zool only.

L 241 assoc of 40Hz calls with season of high prey "first indirect evidence ... food associated calls" how do you square this with Širović et al. 2012: "40-Hz calls peaked in June, preceding a peak in 20-Hz calls by 3-5 months. ... The 40-Hz call may be associated with a foraging function, and temporal separation between 40- and 20-Hz calls may indicate the separation between predominately feeding behavior and other social interactions."?

L 251 should cite more than 31 re seasonality of fin 20 Hz songs in N hemisphere. There is a bigger and stronger literature on this topic.

L 252 Lockyer 1984 is not an appropriate reference for seasonality of breeding of fins in N Atlantic. She emphasizes S oceans for fins and only 1 citation in appendix for N hemisphere, which is N pacific. Best to use primary literature and drop the "assumed" from breeding season.

L 253. I am puzzled at why you cite 49 for long distance communication here but ignore it in lines 161-162 where it directly contradicts your estimated detection range. I think that the longer ranges have stronger evidence, unless you can show by modelling and data that propagation is worse at your site and receiver depth.

L 272 59 argues that singing is LESS common in faster swimmers, the opposite of what this text states

L 303 67 describes a call that was used to coordinate feeding in a fixed group of 8 whales, and 69 to groups of 5-11 whale that were already grouped, not to recruit other whales.

L 310-314 if the 40 Hz calls attract others to feeding opportunity, how can it reduce foraging competition? List the "strong evidence" that the function of the 40 Hz calls is to attract others to feeding opportunity. I know of none.

Lines 320-327 discuss what the consequences would be if the wrong protective measures are taking if the calls do not have the indicated function. Is the current evidence strong enough to make these decisions?

L 331-333 what is the precise evidence that anthropogenic noise overlapping in freq masks detection of finback sounds? What about compensation mechanisms such as increasing source level in increasing noise?

L 336-338 given no evidence for whether or how fins react to noise when making 40 Hz calls, it seems overly speculative to argue it may impact feeding efficiency. What if their reactions compensate for noise? What if noise by itself reduces risk of foraging competition?

L 343 study estimates abundance of prey but cannot estimate availability of prey, if this means rate at which whales can forage, as this depends on much finer patch scales.

Decision letter (RSPB-2021-0288.R0)

12-Apr-2021

Dear Miss Romagosa:

I am writing to inform you that your manuscript RSPB-2021-0288 entitled "Food talk: 40-Hz fin whale calls are associated with prey availability" has, in its current form, been rejected for publication in Proceedings B.

This action has been taken on the advice of referees, who have recommended that substantial revisions are necessary. With this in mind we would be happy to consider a resubmission, provided the comments of the referees are fully addressed. However please note that this is not a provisional acceptance.

Sincerely,
 Dr Daniel Costa
 mailto: proceedingsb@royalsociety.org

Associate Editor
 Comments to Author:
 Dear Authors,

while the study presents interesting data, the manuscript would greatly benefit from streamlining it so the ideas are more clearly presented. Concerns about misrepresentation of the relevance of the findings and statements with claims supported by limited evidence suggest this study is not yet ready for publication and may be a better fit for a more specialized journal.

Overall, I agree with the assessment from the reviewers and hope their specific feedback will help you further improve this work.

Reviewer(s)' Comments to Author:
 Referee: 1
 Comments to the Author(s)
 General

Some rewriting for clarity and conciseness is needed throughout. There are also mis-matched tenses and singular/plural nouns in places. Several long, run on sentences. Much of the paper was overly wordy and presented unneeded details in some places, and missed critical detail in others. In some places wording was awkward. Read through and edit for conciseness and clarity.

Some pertinent details in the data acquisition and analysis are missing

This is modelled prey data tied to automated detected calls. How the data was entered into the model (its format) was missing, and other variables could have been included or current variables refined. All the variables have the possibility of being autocorrelated and so - while I believe the conclusion - it is hard to see how the data presented here is a strong supporter or indicator of the suggested results.

Seems like the authors went into the analysis knowing the answer they expected/wanted - to confirm a singing season and then forage- based calling

A wider literature review is needed - while the authors draw on other species in their introduction, justification of methods and conclusions (which in some cases is ok) they are missing some literature from previous fin whale studies that would inform this study.

Manuscript is long for page limit - and there is a lot of unnecessary material presented
 Supplementary materials not mentioned in text

Abstract

Line 23 – perhaps: Vocalizations in animals show a great variety, with a variety of forms and suggested functions

Line 23- and shows

Line 26 – and whose function – does not read smoothly

Line 29 – mixed tenses

Line 31 – reproductive function

Line 30-32 - Did zooplankton biomass not vary with the season.

Line 33 - Be careful in saying this is the first – there are a number of studies that have suggested the foraging application of the 40-Hz call, and some linking zooplankton measures or proxies to calling. This statement is made throughout.

Line 35 – what do you mean by ‘content’ here. Do you mean structure, or use perhaps

Line 36 – use and not uses

Introduction

Line 44 – reads awkwardly

Line 47 – calls rather than sounds perhaps. I find the authors definition of calls and songs is odd. It may be just purely word choice, but do calls not form song?

Line 48 – are all reproductive displays from males when vocal described as song?

Line 48-49 – these statements need references

Line 50-51 – perhaps some more examples here would help, this is a very simplified and arbitrary distinction of animals that produce song

Line 53 – are songs not comprised by calls? Again the distinction of calls and song is not one I recognise or would support.

Line 56-57, and Line 61 – consider rephrasing

Line 62 – double spacing between ‘individuals.’ and ‘Yet’

Line 65-66 – reads awkwardly

Line 75 – authors say a variety of calls, and then describe just 2, one of which they have previously called a sound.

Line 81 – I believe the references are misused here. Males were most often traveling when calling in reference 21.

Line 82 – counter-calling perhaps rather than call-counter

Line 83 – serves

Line 84 – calls sweep down from 100 Hz to 30 Hz. It is written here in reverse.

Line 88 – there are many studies on the potential function of calls of fin whales not referenced by the authors

Line 89-90 – reads awkwardly.

Line 94 – reference to mating period, but no description of how this is known, or really when this is

Line 99 – double space between megapclicks and).

Methods

As written this is not replicable to this or another area. Some of the justification of model use should have been presented in the introduction

Line 114-115 – gaps in the data are mentioned but not presented, and the reader is simply told it will not have an impact on the analyses

Line 120 – song patterning could still be possible for short periods – and determined if not through the detector through manual analysis. Indeed, Figure 1B clearly shows an example of regularly spaced 20-Hz calls labeled as song and the clip is less than 90 seconds in length.

Line 121 – more description of the methods should be presented here than purely referencing another paper.

122-125 – although it is likely a safe assumption it is an assumption non-the-less, without any verification stated to support it. This section should be this should be better described and clarified. Manual analysis likely help to determine song. Non-song calling has not really been described so far, and is not throughout the paper.

Line 29 – why was manual analysis not done (or at least described in the methods) for 20 Hz calls

122-123 – a small proportion of the automated detected calls

Line 125 – when were the longer duty cycles in comparison to the shorter?

121-126 – was there no manual analysis on the 20-Hz calls – more details of the manual analysis is needed. For example was all data analysed or subsampled? Was this systematically done, guided by the detector

Line 120-126 – did the LFDCS not pick out the presence of 40-Hz calls

Line 135 – be consistent in the way you reference calls (20Hz, 40Hz are written 20-Hz and 40-Hz throughout the rest of the paper).

Line 142 – would be nice to see if/how the model was validated in its expression of zooplankton presence and abundance. Here there is no detail on the studied used to form this model – whether it is spatially or temporally specific, and how many whales/seasons this sampling was completed over. For example, if there are no zooplankton would there be no whales or would they opportunistically eat fish?

Line 151-152 – could some of these physical variables be included rather than 3 variables that essentially represent the same thing.

Line 157-159 – could replace some of the details in 147-148. Rephrase so as a justification of model selection rather than sounding like a result

Line 142 – 159 – much more of this section needs description and justification for the reader

Line 162 – what allows certainty that the whale you are observing is the whale that is vocalising. Were any detection range calculations done to estimate call detection in different ambient noise settings. Detection range is very site dependent - there should be more detail here about how range was estimated/derived and what this range means.

Line 170-171 - I'm not sure, without whale survey data, how this differs from any other time when calls are not heard

Line 171 – is the lack of whales and lack of vocalisations seasonally, if not prey, driven?

Line 174 – you assess collinearity, but establish that there are not patterns in prey driven by season and/or year?

Line 174 – 175 – separate models were built for each call type using the call rate index. Also- why were separate models used?

177 – were these the only 3 variables in the mode? Why would calling differ by year or season alone? How was season defined

178 – details of the interaction variable included is needed – both here and in the results. I assume that zooplankton abundance and season are linked (see also in Figure 2) but at the moment it is unclear how the model will signal that the season is important and not prey or vice versa.

Reference supplementary materials.

185 – AIC – needs to be explained further, it is used in acronym twice. Perhaps these values should also be displayed

191 – this needs a plain language explanation to set in context and better explain to the reader what steps were taken and why

Line 194 – why was a lag of one week used. Do the authors believe that this difference in time would allow any autocorrelation in variables to be rectified, or is this based on a biological phenomenon, or example?

Results

Line 197-198 – consider rewording

Line 197 – is there any control for the number of whales that might be present singing in your rate of calling? There was no whale surveying described in the methods, or anything other than statements of when they were present in the study area or not.

Line 199-201 – I think it should be made clear that this is modelled biomass of prey

Line 202 – lower calling rate

Line 202 – Is this lower than all years – what was the inter-annual variability like? Was it significant between years? The authors have stated that said the interannual seasonal variation was negligible but the annual variation was not?

Figure 2A – 2012 showed the least calling but it and 2011 were the only years to show calling in Jan, Feb. There appears to be patterning between years in call presence/month that is not commented on

Figure 2C is the assumption that the zooplankton biomass is relatively stable in its patterned presence each year?

Figure 2C – the x axis needs a label

Figure 3 – the black dots are not clear, nor is what they represent

Figure 4 – most calls fall outside of the 95% percent confidence area.

Is Figure 2B suggesting that there were not even 2 calls/week in all but one case for 40-Hz calls?

Discussion

Line 241 – again, this is not the first study to have suggested or demonstrated this link or the distinction between call types

Line 243 – could seasonality not be highlighted as a factor of importance because at some periods you have song and others just 20 Hz calls. Perhaps some manual analysis to examine the patterning between calls rather than just number might help this

Line 249-250 – reads awkwardly

Line 256-258 – speculative here. Perhaps wording can make it less so, but the function of the call and the meaning of the calls are 2 very different things

Calling can be more limited when whales are focused on foraging – so type of call may reflect the prey presence, but number or rate may not

The pattern is not consistently seen between years – what was different between, say, 2008 and 2010 – and the other years

- were there any differences between years in the physical/oceanographic variables? What is underlying the differences in prey abundance annually?

2012 – no recovery in the latter season when the zooplankton abundance starts to recover

Line 257 – breeding and feeding occur together? Again is there evidence to support this for your area and times of study

Line 264 – there is the suggestion for humpbacks that calling while feeding is to advertise prey presence and become a more protracted form of courtship – but it doesn't seem to be what you are suggesting here

Line 262-274 – much of this doesn't seem relevant and isn't connected to what has been presented through the paper

Line 275 – 276 – not interannual prey variation? Here this argues too that year and prey are not independent variables.

Line 275 – annual variation mechanism should be described, as should how you know that whale numbers fluctuate

Line 280 – authors say may have affects – did it?

Line 291-292 – this is written as if observations were part of this study, but then reference other studies – rephrase.

307 – the call number seems low even when it is the dominant call source – perhaps more could be detailed in the results about call numbers in this study.

Line 314 – 316 – reads very awkwardly, like there is a word missing

Line 316 – 318 – the authors state what could be possible, but it seems like they had some of this information to hand and did not use it.

Line 325 – if you take into account the detection range.

Line 326-327 – what does different and appropriate mean here.

328 – 338 – not relevant to what is presented here – speculating on aspects of the work that is not examined. Song characteristic were not described at all in this work.

Conclusions – strongly worded and unfounded. Greater literature review needed from the authors before making this statement.

Referee: 2

Comments to the Author(s)

Note that manuscript central makes a mess of text formatting, so I have submitted this as a pdf as well.

This paper makes a real contribution in relating prey abundance to fin whale calling and it deserves to be published. However, there are some problems that require revision. I have corrected textual errors in the annotated pdf and many are listed in the detailed line-by-line comments, but I highlight some here. The text fails to mention some of the most relevant papers on seasonality of 20 and 40 Hz fin calls in the relevant places and needs to include them. The argument on lines 160-161 that fin calls are only detectable at ranges <50km is directly contradicted by the paper cited in the discussion on long range communication. Much better to include both data on Received Levels at different ranges with data on the Source Level of finback calls and transmission loss in your environment and depth of your recorders with respect to sound speed profile to estimate range of detection. It is essential for you to do a better job estimating range of detection and discussing consequences for the scale at which you analyse the data.

I am struck in Figure 4 that the relationship between call rate seems to depend highly on 3-4 points out of a large sample. Any specific info on the context for these? Perhaps worth discussing robustness of result if it is so dependent on a few outliers. I'd be interested to see a scatterplot of call rate by year and season included in the supplementary materials, especially since QAIC for models including these variables was so close to zool only.

Lines 272-314 contain some errors that I flag in the line-by-line

Once these problems addressed in revision, this paper should be ready for publication

Line-by-line comments

line 1 "prey availability" should be changed to "prey abundance" Measuring how available prey is for the whales would require much finer resolution of prey patches

Line 26 The abstract suggests only indirect evidence on functions of fin whale calls was available in past work and that this paper provides direct evidence. This is arguably true for the 40 Hz calls given link to prey here, but the evidence in support of reproductive function of 20 Hz series is no different from that of many past papers and paper should give clearer credit in intro to earlier papers linking 40 Hz calls to foraging season, e.g. Širović et al. 2012. The first line of the conclusion is more accurate.

Line 45: Duration and rhythm do not affect how far a sound can travel nor how well it reaches the receiver (not clear what the difference is between these 2 – are you referring to multi-path?). Longer duration and repeated predictable rhythm may improve ability to detect and classify the call in noise, but that is a different topic.

P49 I do not think most males sing to resolve conflict – it plays a role in male-male competition.

L 53: Females produce song in many species

L 56: what is evidence no calls convey info about the caller as opposed to environment? For example, a threat call may refer to the state of the caller not the environment.

L 65-66: I would argue that you cannot understand the function of a call without also understanding when and how the receiver responds.

L68: The context in which humpbacks sing and how others respond has been studied directly. Not the same for other whales, so does not make sense to lump them. This has been reported for 40Hz calling fin whales, e.g. Wiggins and Hildebrand 2020, so important to include in intro.

l 106 how is depth a proxy for prey and why does it require a time lag?

L 160-161 I think that your justification for detection range of finback calls is very weak, based mainly on unpublished survey data that cannot define detection ranges. detection ranges depend on transmission loss in habitat and depth of recorder. Many sites are predicted to have >200km detection range and in discussion you cite 49 which argues for many hundreds of km range. The short detection range for humpback song is not relevant for finback case. Much better to use data on the SL of finback calls and transmission loss in your environment and depth of your recorders with respect to sound speed profile.

Figure 4. The relationship between call rate seems to depend highly on 3-4 points out of a large sample. Any specific info on the context for these? Perhaps worth discussing robustness of result if it is so dependent on a few outliers. I'd be interested to see scatterplot of call rate by year and season, especially since QAIC was so close to zool only.

L 241 assoc of 40Hz calls with season of high prey "first indirect evidence ... food associated calls" how do you square this with Širović et al. 2012: "40-Hz calls peaked in June, preceding a peak in 20-Hz calls by 3-5 months. ... The 40-Hz call may be associated with a foraging function, and temporal separation between 40- and 20-Hz calls may indicate the separation between predominately feeding behavior and other social interactions."?

L 251 should cite more than 31 re seasonality of fin 20 Hz songs in N hemisphere. There is a bigger and stronger literature on this topic.

L 252 Lockyer 1984 is not an appropriate reference for seasonality of breeding of fins in N Atlantic. She emphasizes S oceans for fins and only 1 citation in appendix for N hemisphere, which is N pacific. Best to use primary literature and drop the "assumed" from breeding season.

L 253. I am puzzled at why you cite 49 for long distance communication here but ignore it in lines 161-162 where it directly contradicts your estimated detection range. I think that the longer ranges have stronger evidence, unless you can show by modelling and data that propagation is worse at your site and receiver depth.

L 272 59 argues that singing is LESS common in faster swimmers, the opposite of what this text states

L 303 67 describes a call that was used to coordinate feeding in a fixed group of 8 whales, and 69 to groups of 5-11 whale that were already grouped, not to recruit other whales.

L 310-314 if the 40 Hz calls attract others to feeding opportunity, how can it reduce foraging competition? List the "strong evidence" that the function of the 40 Hz calls is to attract others to feeding opportunity. I know of none.

Lines 320-327 discuss what the consequences would be if the wrong protective measures are taking if the calls do not have the indicated function. Is the current evidence strong enough to make these decisions?

l 331-333 what is the precise evidence that anthropogenic noise overlapping in freq masks detection of finback sounds? What about compensation mechanisms such as increasing source level in increasing noise?

L 336-338 given no evidence for whether or how fins react to noise when making 40 Hz calls, it seems overly speculative to argue it may impact feeding efficiency. What if their reactions compensate for noise? What if noise by itself reduces risk of foraging competition?

L 343 study estimates abundance of prey but cannot estimate availability of prey, if this means rate at which whales can forage, as this depends on much finer patch scales.

Author's Response to Decision Letter for (RSPB-2021-0288.R0)

See Appendix A.

RSPB-2021-1156.R0

Review form: Reviewer 1

Recommendation

Accept with minor revision (please list in comments)

Scientific importance: Is the manuscript an original and important contribution to its field?

Good

General interest: Is the paper of sufficient general interest?

Good

Quality of the paper: Is the overall quality of the paper suitable?

Good

Is the length of the paper justified?

Yes

Should the paper be seen by a specialist statistical reviewer?

No

Do you have any concerns about statistical analyses in this paper? If so, please specify them explicitly in your report.

No

It is a condition of publication that authors make their supporting data, code and materials available - either as supplementary material or hosted in an external repository. Please rate, if applicable, the supporting data on the following criteria.

Is it accessible?

Yes

Is it clear?

Yes

Is it adequate?

Yes

Do you have any ethical concerns with this paper?

No

Comments to the Author

Thanks to the authors for responding to the previous comments and suggestions. The paper is much improved and is much easier to follow as a reader. The paper is more concise and now presents the necessary information. The increased reference to previous literature helps support the work. Below are a few comments/suggestions for edits. Much more of this work is needed and I am encouraged to see the use of acoustics data being presented in this more ecological framework.

Introduction

Lines 42-45 : feels disjointed, perhaps rephrase the first few sentences, or starting and line 45 "Males of many species.." with 'During reproductive signalling, in many species males...' instead might help make the link to the previous 2 sentences a little easier.

The first paragraph doesn't quite flow

Lines 58-63 in second paragraph are superfluous - you could combine line 63-68 into first paragraph. Or you could just start from Line 69.

Line 79 - why is the '40-Hz call' in quotes here?

Methods

Line 103 - I would think 'sensor' needs capitalising

Line 105 - Try not to start a sentence with an abbreviation

Your description of 40-Hz call on Line 109/Line 130 is inconsistent with your description on Line 79

Line 111-112 - as currently written this is a little confusing as to whether it pertains to your data/study, perhaps reword to make it clear the LFDCS has been used previously in other studies/regions etc.

Line 120-121 - you say here longer duty cycling recording but the reader has no reference yet that the recordings were duty cycled or how

Line 24 - move the reference to the end of the sentence as this refers to blue whale calls

Line 157-138 - resolution of this data? Or is that described line 158-159 - just make sure it is clear

Line 165 - detection range calculation is a result, here a few details of how it was calculated is needed. Reference could made to the supplementary material in the methods, although a paragraph directly stating the calculation would be better. A statement that explains that the call source levels and noise levels used for the calculation were derived the recordings would also be helpful. Some clarification needed on the noise levels used (in Supplementary Materials), were these, for example, the minimum NL for the quietest month and max NL for the nosiest month Same for Line 167-171 - describe this as a sensitivity test of the scale of data integration, and report the results in the next section

You mention the season-prey interaction variable (Line 191) but not any results in the main text. Perhaps just a sentence to state main finding and again refer to Supplementary material.

Line 178-193 - be clear with what are results from this study and what are results from previous work that you are now using

Line 204-208 - suggest rewording

Line 210-211 - lag order of 1 what - this needs a bit more explanation

Results

Figure 4 - and references to it - make sure it is clear that 'observations' is used to indicate call presence

Discussion

Line 278: need a period after 'et al'

Line 282-287 - there are other references that you have made mention too that also support this e.g 43

Line 288: reads a little awkwardly.

Line 292: the difference between 2010 and 2011 is not as striking as you might expect if the result is solely because for whale number/presence

Line 296-299 – No data was presented on this here. Maybe present this a future hypothesis to be tested if not going to show data

Line 307 – careful, reference 43 does not refer to the backscatter intensity directly as copepod biomass but potential prey

Line 322 delete 'foraging competition' to eliminate some of the repetition

References – need to go through, check formatting etc. (e.g. ref 50)

Decision letter (RSPB-2021-1156.R0)

02-Jun-2021

Dear Miss Romagosa

I am pleased to inform you that your manuscript RSPB-2021-1156 entitled "Food talk: 40-Hz fin whale calls are associated with prey biomass" has been accepted for publication in Proceedings B.

The referee(s) have recommended publication, but also suggest some minor revisions to your manuscript. Therefore, I invite you to respond to the referee(s)' comments and revise your manuscript. Because the schedule for publication is very tight, it is a condition of publication that you submit the revised version of your manuscript within 7 days. If you do not think you will be able to meet this date please let us know.

- 1) A text file of the manuscript (doc, txt, rtf or tex), including the references, tables (including captions) and figure captions. Please remove any tracked changes from the text before submission. PDF files are not an accepted format for the "Main Document".
- 2) A separate electronic file of each figure (tiff, EPS or print-quality PDF preferred). The format should be produced directly from original creation package, or original software format. PowerPoint files are not accepted.
- 3) Electronic supplementary material: this should be contained in a separate file and where possible, all ESM should be combined into a single file. All supplementary materials accompanying an accepted article will be treated as in their final form. They will be published alongside the paper on the journal website and posted on the online figshare repository. Files on

figshare will be made available approximately one week before the accompanying article so that the supplementary material can be attributed a unique DOI.

Sincerely,

Dr Daniel Costa

Associate Editor

Comments to Author:

The authors have done a satisfactory job at addressing the concerns brought up. Incorporating the detection range analysis and adding better descriptions to the methodology have greatly improved the manuscript. In addition, more careful attention to the wording and streamlining the introduction make the study easier to follow. There are, however, a few minor points that are unclear or misleading (see list below with line numbers based on "clean" manuscript). The

reviewer also bring up important comments about clarity and flow of the text that deserve attention.

Line 45: If goal is to refer to production of high intensity signals, more clarity is needed.

Line 43: change "cues" to "information". Cues has a specific meaning in animal communication and it is used to describe stimuli whose perception by other animals is not beneficial to the emitter. Using this word in this sentence is misleading.

Line 47-48: what is the second part of the sentence "signal male quality and fitness" adding here? is quality different from body size or health? Also, fitness is unclear and potentially redundant (survival and reproductive ability are different from "quality"?). More clarity is needed here.

Lines 48-57: series of examples about general they provide. Are these sentences just stating that different types of signal provide different information? Without a concluding sentence in this paragraph, it is unclear to the reader what the message is.

Line 58: while it is it easier to perform manipulations in the systems mentioned compared to working with whales, the wording seems to underplay the amount of work required. I suggest adjusting the wording to better reflect the disparity among different study systems in the use of experimental paradigms that allow researchers to test hypotheses about signal function and content.

Lines 69-70: Wording is unclear. is this sentence referring to a comparison to humpback whales?

Lines 285-287: while this is correct, it does not have to necessarily be that way and there are cases in which signals do not provide information about male "quality". Recognizing that further work that directly test that hypothesis is necessary.

Line 301: "fin whale" is twice in this sentence

Line 316: change "cues" to "information"

Lines 352-353: remove "future"

Conclusions: the novelty of the study is heavily based on the methodological approach which overshadows the conceptual contribution. While the methods are worth mentioning, I would suggest emphasizing the biological findings rather than the approach.

Reviewer(s)' Comments to Author:

Referee: 1

Comments to the Author(s).

Thanks to the authors for responding to the previous comments and suggestions. The paper is much improved and is much easier to follow as a reader. The paper is more concise and now presents the necessary information. The increased reference to previous literature helps support the work. Below are a few comments/suggestions for edits. Much more of this work is needed and I am encouraged to see the use of acoustics data being presented in this more ecological framework.

Introduction

Lines 42-45 : feels disjointed, perhaps rephrase the first few sentences, or starting and line 45

"Males of many species..' with 'During reproductive signalling, in many species males...' instead might help make the link to the previous 2 sentences a little easier.

The first paragraph doesn't quite flow

Lines 58-63 in second paragraph are superfluous - you could combine line 63-68 into first paragraph. Or you could just start from Line 69.

Line 79 - why is the '40-Hz call' in quotes here?

Methods

Line 103 - I would think 'sensor' needs capitalising

Line 105 - Try not to start a sentence with an abbreviation

Your description of 40-Hz call on Line 109/Line 130 is inconsistent with your description on Line 79

Line 111-112 – as currently written this is a little confusing as to whether it pertains to your data/study, perhaps reword to make it clear the LFDCS has been used previously in other studies/regions etc.

Line 120-121 – you say here longer duty cycling recording but the reader has no reference yet that the recordings were duty cycled or how

Line 24 – move the reference to the end of the sentence as this refers to blue whale calls

Line 157-138 – resolution of this data? Or is that described line 158-159 – just make sure it is clear

Line 165 – detection range calculation is a result, here a few details of how it was calculated is needed. Reference could made to the supplementary material in the methods, although a paragraph directly stating the calculation would be better. A statement that explains that the call source levels and noise levels used for the calculation were derived the recordings would also be helpful. Some clarification needed on the noise levels used (in Supplementary Materials), were these, for example, the minimum NL for the quietest month and max NL for the nosiest month Same for Line 167-171 – describe this as a sensitivity test of the scale of data integration, and report the results in the next section

You mention the season-prey interaction variable (Line 191) but not any results in the main text. Perhaps just a sentence to state main finding and again refer to Supplementary material.

Line 178-193 – be clear with what are results from this study and what are results from previous work that you are now using

Line 204-208 – suggest rewording

Line 210-211 – lag order of 1 what – this needs a bit more explanation

Results

Figure 4 – and references to it – make sure it is clear that ‘observations’ is used to indicate call presence

Discussion

Line 278: need a period after ‘et al’

Line 282-287 – there are other references that you have made mention too that also support this e.g 43

Line 288: reads a little awkwardly.

Line 292: the difference between 2010 and 2011 is not as striking as you might expect if the result is solely because for whale number/presence

Line 296-299 – No data was presented on this here. Maybe present this a future hypothesis to be tested if not going to show data

Line 307 – careful, reference 43 does not refer to the backscatter intensity directly as copepod biomass but potential prey

Line 322 delete ‘foraging competition’ to eliminate some of the repertition

References – need to go through, check formatting etc. (e.g. ref 50)

Author's Response to Decision Letter for (RSPB-2021-1156.R0)

See Appendix B.

Decision letter (RSPB-2021-1156.R1)

09-Jun-2021

Dear Miss Romagosa

I am pleased to inform you that your manuscript entitled "Food talk: 40-Hz fin whale calls are associated with prey biomass" has been accepted for publication in Proceedings B.

If you are likely to be away from e-mail contact please let us know. Due to rapid publication and an extremely tight schedule, if comments are not received, we may publish the paper as it stands. If you have any queries regarding the production of your final article or the publication date please contact procb_proofs@royalsociety.org

Data Accessibility section

Open Access

Paper charges

Sincerely,

Appendix A

Dear Miss Romagosa:

I am writing to inform you that your manuscript RSPB-2021-0288 entitled "Food talk: 40-Hz fin whale calls are associated with prey availability" has, in its current form, been rejected for publication in Proceedings B.

This action has been taken on the advice of referees, who have recommended that substantial revisions are necessary. With this in mind we would be happy to consider a resubmission, provided the comments of the referees are fully addressed. However please note that this is not a provisional acceptance.

Sincerely,

Dr Daniel Costa
mailto: proceedingsb@royalsociety.org

Associate Editor

Comments to Author:

Dear Authors,

while the study presents interesting data, the manuscript would greatly benefit from streamlining it so the ideas are more clearly presented. Concerns about misrepresentation of the relevance of the findings and statements with claims supported by limited evidence suggest this study is not yet ready for publication and may be a better fit for a more specialized journal.

Overall, I agree with the assessment from the reviewers and hope their specific feedback will help you further improve this work.

Dear Dr. Daniel Costa,

On behalf of my co-authors, I am resubmitting a revised version of the manuscript RSPB-2021-0288 entitled "Food talk: 40-Hz fin whale calls are associated with prey biomass" for consideration as a Proceedings B Research article.

We would like to acknowledge your team's work. The reviewers have given very useful, accurate and constructive criticism and we hope to have matched up your team's work with our improvements to the manuscript. In this version, the topic is more clearly introduced by referencing previous studies, methodology improved to allow replicability and discussion more concisely written based on robust model results. All issues raised by the two reviewers have been fully addressed and main modifications include:

- Wording on the novelty of our results has been adjusted by acknowledging previous studies on fin whale call function and, by focussing on the novel approach used in this study, namely the use of modelled-prey biomass to interpret call function.
- Additional analysis on model robustness has been included to show that the impact of potential influential observations is negligible and hence there is no reason to question the results.
- Detection ranges for both call types have been estimated by considering hydrophone characteristics and deployment locations to justify the spatial scale of modelled prey abundance used. In addition, we compared model-based zooplankton biomass at different spatial scales (centred at the hydrophone positions) to show that changing the extent of spatial analysis won't affect modelling results.
- The methods have been improved with clearer information on analysis procedures and model justification.
- The introduction and discussion have been carefully revised building on the provided comments, streamlining the text while making it more concise and self-explanatory, with all claims being checked for support by either our own results or relevant references.

Authors believe the revised version of this manuscript deserves publication as it demonstrates a clear relationship between a fin whale call type and prey abundance and uses a novel approach to study call function in an elusive marine species. The results of this work are relevant because they set an important baseline for future studies on fin whale vocal behaviour with applicability to conservation.

Please find below our detailed responses to the reviewer's comments.

Reviewer 1

Abstract

Line 23 – perhaps: Vocalizations in animals show a great variety, with a variety of forms and suggested functions.

This sentence has been reworded following the reviewer suggestion.

Line 23- and shows.

This sentence has been reworded following the reviewer suggestion.

Line 26 – and whose function: does not read smoothly.

This sentence has been reworded following the reviewer suggestion.

Line 29 – mixed tenses.

We thank the reviewer for highlighting this error. It has been corrected.

Line 31 – reproductive function.

We thank the reviewer for highlighting this error. It has been corrected.

Line 30-32 - Did zooplankton biomass not vary with the season.

Indeed, zooplankton biomass varied with season with a clear peak in spring and lower values in autumn/winter. To account for the seasonality in zooplankton biomass, the model for the 40-Hz call included an interaction term between season and zooplankton biomass. However, the interaction term was not significant and the best model for the 40-Hz call only included zooplankton biomass. Following the reviewer's comment, we added further details about this interaction and the models in supplementary material (Supplementary Text 1C).

Line 33 - Be careful in saying this is the first – there are a number of studies that have suggested the foraging application of the 40-Hz call, and some linking zooplankton measures or proxies to calling. This statement is made throughout.

We acknowledge that this study is not the first to link 40-Hz calls to prey zooplankton proxies and to suggest a foraging function for this call. However, as far as we know, this is the first study to relate predicted zooplankton biomass to this call type in particular. Following the reviewer's suggestion, we have changed wording throughout the manuscript to properly acknowledge findings from earlier studies and clarify the novelty from this work.

Line 35 – what do you mean by 'content' here. Do you mean structure, or use perhaps.

We meant the information contained in signals. Nevertheless, we acknowledge the term was not clear and could confuse readers and changed the sentence.

Line 36 – use and not uses.

The sentence was reworded.

Introduction

Some of the justification of model used should be given in the introduction.

We agree with the reviewer that the manuscript did not put enough emphasis on model justification. To address this issue, we tried to briefly explain why it was necessary to use a zooplankton model by rewording the last paragraph of the introduction to: "In the absence of concurrent measurements of prey biomass, an ecosystem model was used to provide hindcast simulations of low trophic level (mesozooplankton) biomass for the area and period of acoustic recordings. This approach allowed investigating the direct relationship between fin whale vocal behaviour and predicted prey biomass, avoiding interpretation of relationships with time lagged prey proxies (i.e., chlorophyll)."

Line 44 – reads awkwardly.

We have changed the first three paragraphs of the introduction, including this section.

Line 47 – calls rather than sounds perhaps. I find the authors definition of calls and songs. is odd. It may be just purely word choice, but do calls not form song?

We acknowledge that distinction between calls and songs was confusing in the manuscript. Given that literature do not provide a clear definition of these two terms, we chose to use the term call for both vocalisation types, and just mention in methods that 20-Hz calls are part of songs.

Line 48 – are all reproductive displays from males when vocal described as song?

The reviewer makes an interesting point here. In fact, some animals do produce reproductive calls that are not considered songs, like male deer vocalisations. We have changed the first three paragraphs of the introduction, including this section.

Line 48-49 – these statements need references.

We have changed the first three paragraphs of the introduction, including this section.

Line 50-51 – perhaps some more examples here would help, this is a very simplified and arbitrary distinction of animals that produce song

We have changed the first three paragraphs of the introduction, including this section.

Line 53 – are songs not comprised by calls? Again the distinction of calls and song is not one I recognise or would support.

As said in one of the comments above, authors acknowledge that distinction between calls and songs was confusing in the manuscript and decided to simplify by calling both vocalisations calls.

Line 56-57, and Line 61 consider rephrasing.

We have changed the first three paragraphs of the introduction, including this section.

Line 62 – double spacing between 'individuals.' and 'Yet'.

We have changed the first three paragraphs of the introduction, including this section.

Line 65-66 – reads awkwardly.

We have changed the first three paragraphs of the introduction, including this section.

Line 75 – authors say a variety of calls, and then describe just 2, one of which they have previously called a sound.

This sentence has been reworded.

Line 81 – I believe the references are misused here. Males were most often traveling when calling in reference 21.

The Reviewer probably misidentified the reference. Croll et al. (2002) suggest that fin whale song may be used by males to advertise food to females. Here, we refer to this statement.

Line 82 – counter-calling perhaps rather than call-counter.

We thank the reviewer for highlighting this mistake.

Line 83 – serves.

Here we refer to both call types so we used the plural.

Line 84 – calls sweep down from 100 Hz to 30 Hz. It is written here in reverse.

We thank the reviewer for highlighting this mistake.

Line 88 – there are many studies on the potential function of calls of fin whales not referenced by the authors.

We acknowledge the fairness of the criticism and added appropriate references for both call types in the introduction and discussion. Also, we removed the sentence stating “limited knowledge”.

Line 89-90 – reads awkwardly.

We have reworded this paragraph.

Line 94 – reference to mating period, but no description of how this is known, or really when this is.

The timing of fin whale mating is mentioned in the previous paragraph, with the appropriate references.

Line 99 – double space between megapclicks and).

It has been corrected.

Methods

Line 114-115: gaps in the data are mentioned but not presented, and the reader is simply told it will not have an impact on the analyses.

We acknowledge that gaps in the dataset were not clearly reported in the manuscript and were only shown in supplementary material. To address this, we added a graph on the x-axis of figure 2 to illustrate gaps by month and year.

Line 120: song patterning could still be possible for short periods – and determined if not through the detector through manual analysis. Indeed, Figure 1B clearly shows an example of regularly spaced 20-Hz calls labelled as song and the clip is less than 90 seconds in length.

We completely agree with the reviewer and replaced the sentence with more information on the manual analysis to distinguish song from non-song 20-Hz calls.

Line 121: more description of the methods should be presented here than purely referencing another paper.

We acknowledge that information on the methodology was scarce and added the most important details of the automatic detection.

Line 122-125: although it is likely a safe assumption it is an assumption non-the-less, without any verification stated to support it. This section should be this should be better described and clarified. Manual analysis likely help to determine song. Non-song calling has not really been described so far, and is not throughout the paper.

We thank the reviewer for his/her comment. To address it, we sampled one month of recordings with longer duty cycles per season. Results showed that only 2.5% of files contained non-song 20-Hz notes (Oct: 0%; Nov: 3.5% and March:0%). We included these results in the manuscript in support of our assumption.

Line 29: why was manual analysis not done (or at least described in the methods) for 20 Hz calls.

20-Hz calls were far more common in our dataset than 40-Hz calls. An automatic detector was used to facilitate the detection of 20-Hz calls in the entire dataset. The detector was used as part of a previous study and results used for this study. We reworded this section to clarify the process of call detection and analysis.

Line 122-123: a small proportion of the automated detected calls.

The paragraph has been modified.

Line 125 – when were the longer duty cycles in comparison to the shorter?

Longer duty cycles were used in 2011/2012, as presented in supplementary Figure S1. To reduce potential bias caused by different duty cycles used in this study, a call rate index was calculated by

dividing number of calls per sampling time, in hours, during that week. This is explained in the last paragraph of the method section (a) Acoustic data collection and analyses.

Line 121-126: was there no manual analysis on the 20-Hz calls – more details of the manual analysis is needed. For example was all data analysed or subsampled? Was this systematically done, guided by the detector.

An automatic analysis was performed for the 20-Hz call (now better explained) and a manual analysis for the 40-Hz call, and all data were analysed. We recognize this section was confusing and we have changed it for clarity.

Line 120-126: did the LFDCS not pick out the presence of 40-Hz calls.

Identification of 40-Hz calls using automatic detectors is challenging because of the frequency overlap with other baleen whale calls, like those from sei or blue whales. So, 40-Hz calls were detected manually by visually inspecting spectrograms for the entire dataset. We included this explanation in the methods section of the manuscript.

Line 135: be consistent in the way you reference calls (20Hz, 40Hz are written 20-Hz and 40-Hz throughout the rest of the paper).

We thank the reviewer for this comment and changed it throughout the manuscript.

Line 142 – 159: much more of this section needs description and justification for the reader.

More information on the model validation has been added as well as clarification about the temporal and spatial scales used matching the acoustic recordings.

Line 142: would be nice to see if/how the model was validated in its expression of zooplankton presence and abundance. Here there is no detail on the studied used to form this model – whether it is spatially or temporally specific, and how many whales/seasons this sampling was completed over. For example, if there are no zooplankton would there be no whales or would they opportunistically eat fish?

We added more information on the validation of the zooplankton model and on the model itself. We explain that model validation was done by comparing model predictions with the climatological database COPEPOD that provides standardised mean zooplankton biomass values on a global spatial grid (Masina et al., 2017; von Schuckmann et al., 2020). We also explained that the SEAPODYM model provides spatially explicit (resolution is $0.25^\circ \times 0.25^\circ$) weekly estimates of zooplankton biomass, and that these were obtained for the entire period matching the acoustic recordings.

The dataset used in this study does not enable assessing if fin whales would eat fish in the absence of zooplankton or if they would instead leave the area. The question raised by the reviewer does not in any way alter the finding that 40-Hz calls were positively associated with modelled zooplankton biomass, giving support to the previous suggestion that the call is used in a feeding context. On the other hand, it could be argued that we did not find a relationship between production of 20-Hz calls with zooplankton biomass because fin whales were foraging on fish. We find this highly unlikely. First, because although less abundant than in spring, our results show that zooplankton was still present during late autumn and winter when 20-Hz calls peaked. Second, because in a previous study combining satellite tracking data and SEAPODYM based estimates of mesozooplankton and

micronekton (6 functional groups: epipelagic, migrant upper mesopelagic, upper mesopelagic, highly migrant lower mesopelagic, migrant lower mesopelagic and lower mesopelagic), only mesozooplankton was found to be important at explaining fin whale distribution in the Azores and across the mid-North Atlantic Ocean (Pérez-jorge et al., 2020). We showed that fin whale migration through the Azores was closely linked to the availability of estimates of zooplankton here. As zooplankton became scarcer in the Azores during late spring and early summer, predicted distribution of fin whales shifted northwards (Pérez-jorge et al., 2020). These results are in agreement with the seasonality of sightings of fin whales (Silva et al., 2014).

To address the concerns raised by the reviewer, we added some findings of our previous study (Pérez-jorge et al., 2020) to reinforce our assumption that fin whales in the study area primarily feed on zooplankton.

Line 151-152: could some of these physical variables be included rather than 3 variables that essentially represent the same thing.

Physical variables showed autocorrelation with zooplankton biomass so were not included in the model. Furthermore, the aim of the study was to understand the relationship of call rates with prey biomass directly, as opposed to proxies of prey, which then require other assumptions to interpret time-lagged relationships.

Line 157-159: could replace some of the details in 147-148. Rephrase so as a justification of model selection rather than sounding like a result.

This sentence was included as a justification of the model, as the reviewer suggests as such “In addition, mesozooplankton biomass derived from a spatial ecosystem and population dynamics model (SEAPODYM) was the most important predictor of the distribution of fin whales in the Azores and across the mid-North Atlantic, whilst micronekton biomass estimates from the same model had no effect on the movements of the species (Pérez-jorge et al., 2020). Thus, we assumed that zooplankton is the main prey of fin whales in the study area and obtained estimates of zooplankton biomass from the lower trophic level.”

Line 162: what allows certainty that the whale you are observing is the whale that is vocalising. Were any detection range calculations done to estimate call detection in different ambient noise settings. Detection range is very site dependent - there should be more detail here about how range was estimated/derived and what this range means.

We completely agree with the reviewer. We have calculated detection ranges for the hydrophones (EARs) used, for each deployment location and different noise levels recorded in the area. All this information has been included in supplementary material (Supplementary Fig. S1). In addition, we compared model-based zooplankton biomass at different spatial scales (centred at the hydrophone positions) to show that changing the extent of spatial analysis will not affect modelling results (Supplementary Fig. S2).

Line 170-171: I'm not sure, without whale survey data, how this differs from any other time when calls are not heard. Line 171: is the lack of whales and lack of vocalisations seasonally, if not prey, driven.

The aim of the present study was to understand fin whale vocal behaviour when animals are present in the area, and not what drives their presence or abundance. Sighting records have shown that fin whales rarely occur in the study area in the summer (Silva et al., 2014; Visser et al., 2011) and that prey availability or environmental variables linked to prey availability drive the fin whale northward migration and their disappearance from the area in summer months (Pérez-jorge et al., 2020; Prieto et al., 2017; Visser et al., 2011). Thus, using summer months in the models, when animals are known to be absent in the area, would only confound interpretation of the temporal patterns in vocal behaviour, as the data would not allow distinguishing vocal changes from presence/absence of whales.

Line 174: you assess collinearity, but establish that there are not patterns in prey driven by season and/or year?

We acknowledge this subject was not properly addressed in the previous version and added a paragraph in the text as follows: “The variance inflation factor (VIF) was calculated for the complete models to measure the strength of correlation between all predictor variables (season, year and zooplankton biomass). Values higher than 5 or 10 are considered too high and could cause misinterpretation of model outputs (Montgomery and Peck, 1992). In our models, VIF values for the three variables were approximately one.”

Line 174 – 175: separate models were built for each call type using the call rate index. Also- why were separate models used?

We used two separate models because we were interested in understanding the effects of the same predictor variables on the calling rates of each call type. We recognize the justification for separate models was missing in the manuscript and was now added to the text.

Line 177: were these the only 3 variables in the model? Why would calling differ by year or season alone? How was season defined?

The full model indeed included zooplankton biomass, season, year and an interaction term between zooplankton biomass and season. These variables were included because we were specifically interested in testing the hypotheses that 40-Hz calls are a food-associated call, and that 20-Hz calls are used to attract females via food advertising. These predictions were described in the last paragraph of the Introduction. Because our focus was on zooplankton biomass, we did not include prey proxies in the models (in addition, as explained before, these were strongly correlated with zooplankton biomass).

Season and year were included in the model to assess how production of each call type varied among years and seasons. The rationale for the use of these variables was also explained in the last paragraph of the Introduction.

We improved the explanation of the definition of seasons to make it clearer.

Line 178: details of the interaction variable included is needed – both here and in the results. I assume that zooplankton abundance and season are linked (see also in Figure 2) but at the moment it is unclear how the model will signal that the season is important and not prey or vice versa. Reference supplementary materials.

An interaction term between these two variables was included in the models because season could influence the relationship between call rates and modelled zooplankton biomass. Models for each call type were built using all possible combinations of the explanatory variables and the goodness-of-fit was assessed using the Quasi-Akaike's Information Criterion. The section on the statistical model selection and validation was changed to address some of the comments and questions.

Line 185: AIC – needs to be explained further, it is used in acronym twice. Perhaps these values should also be displayed.

We thank the reviewer for this comment. We spelled out the name of QAIC (Quasi-Akaike's Information Criterion), and added more information about QAIC calculation in supplementary text 1C. Values of QAIC for all models are presented in supplementary table S2. Two variants of QAIC are also displayed in table S2, namely: $\Delta(\text{AIC})$ and Weight (AIC). Descriptions of these two measures are shown in the footnote of supplementary table S2.

Line 191: this needs a plain language explanation to set in context and better explain to the reader what steps were taken and why.

This paragraph has been reworded and reasons for each check plot added.

Line 194: why was a lag of one week used. Do the authors believe that this difference in time would allow any autocorrelation in variables to be rectified, or is this based on a biological phenomenon, or example?

We used a lag of one week for the response variable to correct for autocorrelation. After adding the response variable with a lag of order one to the set of predictors, we re-estimated the model and analysed the residuals. The autocorrelation function and the partial autocorrelation function of the residuals did not show any autocorrelation. Thus, from a statistical point of view, there was no need to include more lagged values of the response variable in the model. We acknowledged this was poorly explained in the previous version of the manuscript and added a sentence to the last paragraph of the "Methods" clarifying the use of the lagged response variable.

Results

Line 197-198: consider rewording.

The sentence has been reworded.

Line 197: is there any control for the number of whales that might be present singing in your rate of calling? There was no whale surveying described in the methods, or anything other than statements of when they were present in the study area or not.

Unfortunately, we have no way of assessing the number of calling whales in the area. Although we agree this information would have assisted with the interpretation of the data, we argue that its absence does not invalidate our interpretation of the results. As we stated before, we were not interested in using acoustic detections to investigate whale presence or abundance but to assess changes in vocal behaviour when whales were present and contribute to understanding the reasons underlying these changes. We would also like to stress that our study covered different years and the patterns in fin whale vocal behaviour were highly consistent between years.

Line 199-201: I think it should be made clear that this is modelled biomass of prey.

The sentence was changed accordingly.

Line 202: lower calling rate.

The sentence was changed accordingly.

Line 202: Is this lower than all years – what was the inter-annual variability like? Was it significant between years? The authors have stated that said the interannual seasonal variation was negligible but the annual variation was not?

Inter-annual variability for both call types was assessed with the models by including year as an explanatory variable. Best model for the 20-Hz call rates showed that year was an important predictor, while the best model for the 40-Hz call did not include year. Hence, we only discuss inter-annual variability for the 20-Hz call. Nevertheless, we acknowledge that it was not appropriate to talk about inter-annual variability before explaining model outputs and decided to remove this paragraph from this section.

Figure 2A: 2012 showed the least calling but it and 2011 were the only years to show calling in Jan, Feb. There appears to be patterning between years in call presence/month that is not commented on.

We apologise for not reporting gaps in the dataset with clarity and consequently added a graph in the x-axis of Figure 2 with sampling effort by year and month. Available data for January and February is for years 2011 and 2012. Other years do not have data for these months.

Is Figure 2B suggesting that there were not even 2 calls/week in all but one case for 40-Hz calls?

Call rates in the figure and in models refer to call rate index, that is number of calls divided by sampling time to correct for different duty cycles used in the study.

Figure 2C: is the assumption that the zooplankton biomass is relatively stable in its patterned presence each year?

The seasonal pattern in zooplankton biomass was indeed consistent across years, although investigating the reasons for this pattern is beyond the scope of this manuscript.

Figure 2C: the x axis needs a label

The label was added.

Figure 3: the black dots are not clear, nor is what they represent.

Size of black dots has been increased for better visualisation and their description improved in the figure legend.

Figure 4: most calls fall outside of the 95% percent confidence area.

Figure 4 displays the relationship between zooplankton biomass and the 40-Hz call rate index in terms of the overall trend. More precisely, when fitting the quasi-Poisson model to the dataset, we obtained the quasi-maximum likelihood estimate for the underlying model's mean and variance. To summarize the estimated model's most essential features to describe the 40-Hz call rate index, we plotted the estimates for the mean and the respective 95% confidence intervals. Thus, Figure 4 shows the overall trend (i.e. mean) of the estimated model.

To include this relevant comment in the text, we rewrote the legend of Fig. 4: "Zooplankton biomass effect on the 40-Hz call. Blue points represent observations. The solid line corresponds to the fitted mean value (that is, the trend); the grey shaded areas represent the 95% confidence intervals."

Also, we added more information about models' goodness-of-fit in the text (for both 20-Hz call rate and 40-Hz call rate), namely the half-normal plot of the Pearson residuals with simulated envelope (based on 1000 runs). These graphs also allow checking for outliers. From our perspective, the graphs will allow the reader to evaluate the main features of the fitted models without entering into the theoretical details of the underlying statistical models. We added information about the half-normal plots in the "Methods" section; in the "Results" section, we wrote two sentences about the results from the application of this tool to our datasets (20-Hz call rate and 40-Hz). Plots are shown in supplementary material (Supplementary Figs. S3 and S4).

Discussion

Line 241: Again, this is not the first study to have suggested or demonstrated this link or the distinction between call types.

We agree that this study is not the first to suggest a food-associated function for this call or to use zooplankton or proxies in relation to call activity. However, as far as we know, it is the first study to relate modelled prey data specifically to 40-Hz calls. We changed the wording to clarify this novelty.

Line 243: Could seasonality not be highlighted as a factor of importance because at some periods you have song and others just 20 Hz calls. Perhaps some manual analysis to examine the patterning between calls rather than just number might help this.

Manual analysis detected only 2.5 % of non-song 20-Hz calls in longer duty cycle recordings in three months representative of each season. Hence, we can safely assume that non-song 20-Hz notes represent a very small percentage in our dataset and have only a very minor effect on the seasonal pattern of 20-Hz notes reported in this study.

Line 249-250: reads awkwardly.

The sentence was reworded.

**Line 256-258: speculative here. Perhaps wording can make it less so, but the function of the call and the meaning of the calls are 2 very different things
Calling can be more limited when whales are focused on foraging – so type of call may reflect the prey presence, but number or rate may not
The pattern is not consistently seen between years – what was different between, say, 2008 and 2010 – and the other years
- were there any differences between years in the physical/oceanographic variables? What is**

underlying the differences in prey abundance annually? – 2012 – no recovery in the latter season when the zooplankton abundance starts to recover.

We are not sure to have understood the reviewer's comment correctly and we did our best to change this section of the manuscript to address the comment adequately. The sentence in Line 256-257 refers to Croll et al., (2002) suggestion that male 20-Hz song would serve to attract females to aggregations of patchily distributed prey. We argue that our findings do not support this hypothesis, as we failed to find any relationship between 20-Hz call rates and zooplankton biomass. The reviewer states that "type of call may reflect the prey presence, but number or rate may not". As we note in the Discussion with respect to the 40-Hz call rates, there are few examples in the literature of mammals or birds producing acoustically distinctive calls in feeding context, or using acoustically distinctive calls to signal food type, quality or quantity (Clay et al., 2012). Instead, it seems that this information may be conveyed (if conveyed at all) through different acoustic features within a call type, namely call rate (Clay et al., 2012). We made extensive changes in the Discussion to clarify this point, drawing from what is currently known about food-associated calls in birds and mammals in general.

Line 257: breeding and feeding occur together? Again is there evidence to support this for your area and times of study.

Again, this sentence referred to the suggestion made by Croll et al., (2002), not to our study area where breeding has never been demonstrated. We reworded this entire section in an attempt to clarify our reasoning.

Line 264: there is the suggestion for humpbacks that calling while feeding is to advertise prey presence and become a more protracted form of courtship – but it doesn't seem to be what you are suggesting here.

The discussion has been modified and this sentence deleted.

Line 262-274: much of this doesn't seem relevant and isn't connected to what has been presented through the paper –

We agree with the reviewer and removed much of this section. However, we kept the first sentence which we believe provides the most plausible explanation for our results with respect to the 20-Hz song of male fin whales, its ultimate function and the underlying mechanism.

Line 275 – 276: not interannual prey variation? Here this argues too that year and prey are not independent variables.

We changed this last paragraph to better explain our interpretation of the results.

Line 275: annual variation mechanism should be described, as should how you know that whale numbers fluctuate.

We agree with the reviewer that this section required a better description. We added visual data collected by the Azores Fisheries Observer program that confirm that 2012 less fin whales were sighted compared to other years which further suggest that lower call rates reflect lower number of whales.

Line 280: authors say may have affects – did it?

The paragraph has been modified.

Line 291-292: this is written as if observations were part of this study, but then reference other studies – rephrase.

This was rephrased to address this comment.

Line 307: the call number seems low even when it is the dominant call source – perhaps more could be detailed in the results about call numbers in this study.

Call rates refer to call rate index, which is number of calls divided by sampling period to correct for the different duty cycles used in this study. This causes call rates to have much lower numbers.

Line 314 – 316: reads very awkwardly, like there is a word missing –

This entire section has been modified.

Line 316 – 318: the authors state what could be possible, but it seems like they had some of this information to hand and did not use it.

This entire section has been modified.

Line 325 – if you take into account the detection range.

The section “Implications for conservation” has been deleted and its most relevant information included at the end of the conclusion.

Line 326-327: what does different and appropriate mean here.

The section “Implications for conservation” has been deleted and its most relevant information included at the end of the conclusion as such: “Understanding the function of animal vocalisations can help identifying functional habitats and support conservation planning (Teixeira et al., 2019). Information on the temporal and spatial occurrence of fin whale 40-Hz calls may inform when and where animals engage in foraging and provide important clues to the environmental factors promoting foraging behaviour on this species. Similarly, the 20-Hz song may give unique insights into the location and characteristics of the areas used for mating. Studies combining visual and acoustic observations of callers and receivers simultaneously, offering information on the behavioural context of call production along with responses of conspecifics, could significantly advance our understanding of fin whale vocal behaviour”

Line 328 – 338: not relevant to what is presented here – speculating on aspects of the work that is not examined. Song characteristic were not described at all in this work.

This paragraph has been removed following the reviewer suggestion.

Reviewer 2:

Abstract

Line 1 - "prey availability" should be changed to "prey abundance" Measuring how available prey is for the whales would require much finer resolution of prey patches.

Authors agree with the reviewer and have changed "prey availability" to "prey biomass" throughout the manuscript.

Line 26 - The abstract suggests only indirect evidence on functions of fin whale calls was available in past work and that this paper provides direct evidence. This is arguably true for the 40 Hz calls given link to prey here, but the evidence in support of reproductive function of 20 Hz series is no different from that of many past papers and paper should give clearer credit in intro to earlier papers linking 40 Hz calls to foraging season, e.g. Širović et al. 2012. The first line of the conclusion is more accurate.

We acknowledge that our choice of words was unfortunate and that more credit should have been given to previous studies on fin whale 40-Hz call. We have removed "indirect evidence" and have added more references to previous studies on 40-Hz call activity in the introduction and discussion.

Introduction

Line 45 - Duration and rhythm do not affect how far a sound can travel nor how well it reaches the receiver (not clear what the difference is between these 2 – are you referring to multi-path?). Longer duration and repeated predictable rhythm may improve ability to detect and classify the call in noise, but that is a different topic.

We completely agree with the reviewer comment. We have made substantial changes to this part of the Introduction and removed this entire sentence.

Line 49 - I do not think most males sing to resolve conflict – it plays a role in male-male competition.

We completely agree with the reviewer comment. We have made substantial changes to this part of the Introduction. This sentence is now: "Males of many species sing loudly and more intensely during the mating season to attract or court females, repel conspecific males, or both".

Line 53: Females produce song in many species.

Once again, we agree with the reviewer. We have made substantial changes to this part of the Introduction and this sentence was removed.

Line 56: what is evidence no calls convey info about the caller as opposed to environment? For example, a threat call may refer to the state of the caller not the environment.

Once again, we agree with the reviewer. We have made substantial changes to this part of the Introduction. This sentence is now: "Many species give alarm calls to inform conspecifics about a threat (Dezecache and Berthet, 2018). The propensity to emit alarm calls can vary with predator type and abundance (Thorley and Clutton-Brock, 2017), as well as with the signaller's stress, perception of risk, social ranking and relationships (Nash et al., 2021)".

Line 65-66: I would argue that you cannot understand the function of a call without also understanding when and how the receiver responds.

We agree with the reviewer that our wording was not concise enough. We have made substantial changes to this part of the Introduction and now the paragraph reads as: “The notable exception are the vocalizations of humpback whales (*Megaptera novaengliae*), several of which have been linked to foraging activities (Cerchio and Dahlheim, 2001; Stimpert et al., 2007) or to social interactions (Dunlop et al., 2007; Noad et al., 2006). Similarly, observations of the social and behavioural context in which male humpback whales sing, and of the response of conspecifics, support the dual role of humpback whale song in female attraction (via lone singers or communal singing), and in mediating interactions among males (Cholewiak et al., 2018; Herman, 2016; Smith et al., 2008)”.

Line 68: The context in which humpbacks sing and how others respond has been studied directly. Not the same for other whales, so does not make sense to lump them. This has been reported for 40Hz calling fin whales, e.g. Wiggins and Hildebrand 2020, so important to include in intro.

We have changed the first three paragraphs of the introduction and state that playback experiments have been conducted in humpback whales. References to Wiggins and Hildebrand (2020) work have been added in the introduction and discussion.

Line 106 how is depth a proxy for prey and why does it require a time lag?

We thank the reviewer for pointing out this mistake. We meant to say chlorophyll and changed the sentence accordingly.

Methods

Line 160-161: I think that your justification for detection range of finback calls is very weak, based mainly on unpublished survey data that cannot define detection ranges. detection ranges depend on transmission loss in habitat and depth of recorder. Many sites are predicted to have >200km detection range and in discussion you cite 49 which argues for many hundreds of km range. The short detection range for humpback song is not relevant for finback case. Much better to use data on the SL of finback calls and transmission loss in your environment and depth of your recorders with respect to sound speed profile.

We completely agree with the reviewer and have calculated detection ranges for our hydrophones (EARs), at each deployment location and different noise levels recorded in the area. All this information has been included in supplementary material (Supplementary Text 1b). In addition, we compared model-based zooplankton biomass at different spatial scales (centred at the hydrophone positions) to show that changing the extent of the spatial analysis won't affect modelling results (Supplementary Fig. S2).

Results

Figure 4. The relationship between call rate seems to depend highly on 3-4 points out of a large sample. Any specific info on the context for these? Perhaps worth discussing robustness of result if it is so dependent on a few outliers. I'd be interested to see scatterplot of call rate by year and season, especially since QAIC was so close to zool only.

The reviewer raises an important question that needs to be addressed and clarified. According to the model validation and performance testing, the 3-4 points noted by the reviewer were not identified as outliers. Please see response to reviewer 1 comment “Figure 4: most calls fall outside of the 95% percent confidence area”, where details of check plots for outliers are described.

In terms of the robustness of the models, we applied the Quasi-Poisson model, which, to a certain extent, can be considered a robust procedure. When dealing with this type of models, we do not need to assume the mathematical expression for the density function of the random variable under consideration. Instead, the quasi-likelihood methods solely rely on the relationship between the mean and the variance of the distribution, which, in this case, also accommodates an overdispersion parameter. The quasi-maximum likelihood estimators are consistent and asymptotically Normal. Standard statistical inference developed in the likelihood settings can be easily generalized to the quasi-likelihood framework (Wedderburn, 1974). Therefore, this methodology can be considered robust to the existence of outliers.

Scatterplot of call rates versus season and year show that 40-Hz call rate is not as seasonally and inter-annually variable as the 20-Hz call rate. This small variability in the 40-Hz call is shadowed by the strong effect of zooplankton biomass on call rates and this is why the best model only includes zooplankton biomass.

Discussion

Line 241: Assoc of 40Hz calls with season of high prey “first indirect evidence ... food associated calls” how do you square this with Širović et al. 2012: “40-Hz calls peaked in June, preceding a peak in 20-Hz calls by 3–5 months. ... The 40-Hz call may be associated with a foraging function, and temporal separation between 40- and 20-Hz calls may indicate the separation between predominately feeding behavior and other social interactions.”?

We completely agree with the reviewer that this study is not the first in suggesting a foraging function for this call. However, as far as we know, it is the first study in relating modelled prey data to this call type in particular. We changed the wording for clarity.

Line 251: should cite more than 31 re seasonality of fin 20 Hz songs in N hemisphere. There is a bigger and stronger literature on this topic.

We agree with the reviewer and added more references.

Line 252: Lockyer 1984 is not an appropriate reference for seasonality of breeding of fins in N Atlantic. She emphasizes S oceans for fins and only 1 citation in appendix for N hemisphere, which is N pacific. Best to use primary literature and drop the “assumed” from breeding season.

The reviewer is right and we have added the following references:

- Ohsumi S, Nishiwaki M, Hibiya T. 1958 Growth of fin whale in the Northern Pacific. Sci. Reports Whales Res. Inst. 13, 97–133.

- Kjeld MJ. 1992 Sex hormone concentrations in blood serum from the north Atlantic fin whale (*Balaenoptera physalus*). J. Endocrinol. 134, 405–413.

Line 253: I am puzzled at why you cite 49 for long distance communication here but ignore it in lines 161-162 where it directly contradicts your estimated detection range. I think that the longer ranges have stronger evidence, unless you can show by modelling and data that propagation is worse at your site and receiver depth.

As explained above, we calculated the detection ranges for our hydrophones EARs at the deployment locations (See supplementary material for more details). Detection ranges of our hydrophones were lower than values reported in this reference.

Line 272: 59 argues that singing is LESS common in faster swimmers, the opposite of what this text states.

This was what we were trying to say but our wording wasn't clear. Nevertheless, this paragraph has been removed as one reviewer suggested this was not relevant for this study.

Line 303: 67 describes a call that was used to coordinate feeding in a fixed group of 8 whales, and 69 to groups of 5-11 whale that were already grouped, not to recruit other whales.

We acknowledge this was poorly worded and replaced “recruit” by cooperative behaviour.

Lines 310-314: if the 40 Hz calls attract others to feeding opportunity, how can it reduce foraging competition? List the “strong evidence” that the function of the 40 Hz calls is to attract others to feeding opportunity. I know of none.

We acknowledge the wording was confusing. The whole paragraph has been modified for clarity.

Lines 320-327: discuss what the consequences would be if the wrong protective measures are taking if the calls do not have the indicated function. Is the current evidence strong enough to make these decisions?

The section “Implications for conservation” has been deleted and its most relevant information included at the end of the conclusion.

Lines 331-333: what is the precise evidence that anthropogenic noise overlapping in freq masks detection of finback sounds? What about compensation mechanisms such as increasing source level in increasing noise?

The whole paragraph has been deleted as suggested by one of the reviewers.

Line 336-338: given no evidence for whether or how fins react to noise when making 40 Hz calls, it seems overly speculative to argue it may impact feeding efficiency. What if their reactions compensate for noise? What if noise by itself reduces risk of foraging competition?

The whole paragraph has been deleted as suggested by one of the reviewers.

Line 343: study estimates abundance of prey but cannot estimate availability of prey, if this means rate at which whales can forage, as this depends on much finer patch scales.

We have changed “availability” to “biomass” throughout the manuscript as the reviewer suggests.

REFERENCES

- Cerchio, S., and Dahlheim, M. (2001). “Variation in feeding vocalizations of humpback whales megaptera novaeangliae from southeast alaska,” *Bioacoustics*, **11**, 277–295. doi:10.1080/09524622.2001.9753468
- Cholewiak, D. M., Cerchio, S., Jacobsen, J. K., Urbán-R., J., and Clark, C. W. (2018). “Songbird dynamics under the sea: Acoustic interactions between humpback whales suggest song mediates male interactions,” *R. Soc. Open Sci.*, **5**, 171298. doi:10.1098/rsos.171298
- Clay, Z., Smith, C. L., and Blumstein, D. T. (2012). “Food-associated vocalizations in mammals and birds: What do these calls really mean?,” *Anim. Behav.*, **83**, 323–330. doi:10.1016/j.anbehav.2011.12.008
- Croll, A. D., Clark, C. W., Acevedo, A., Tershy, B., Flores, S., Gedamke, J., and Urban, J. (2002). “Only male fin whales sing loud songs,” *Nature*, **417**, 5–7.
- Dezecache, G., and Berthet, M. (2018). “Working hypotheses on the meaning of general alarm calls,” *Anim. Behav.*, **142**, 113–118. doi:10.1016/j.anbehav.2018.06.008
- Dunlop, R. A., Noad, M. J., Cato, D. H., and Stokes, D. (2007). “The social vocalization repertoire of east Australian migrating humpback whales (*Megaptera novaeangliae*),” *J. Acoust. Soc. Am.*, **122**, 2893. doi:10.1121/1.2783115
- Herman, L. M. (2016). “The multiple functions of male song within the humpback whale (*Megaptera novaeangliae*) mating system: review, evaluation, and synthesis,” *Biol. Rev.*, , doi: 10.1111/brv.12309. doi:10.1111/brv.12309
- Masina, S., Storto, A., Ferry, N., Valdivieso, M., Haines, K., Balmaseda, M., Zuo, H., et al. (2017). “An ensemble of eddy-permitting global ocean reanalyses from the MyOcean project,” *Clim. Dyn.*, **49**, 813–841. doi:10.1007/s00382-015-2728-5
- Montgomery, D. C., and Peck, E. A. (1992). *Introduction to linear regression analysis*, Willey, New York, 504 pages.
- Nash, A. L., Jebb, A. H. M., and Blumstein, D. T. (2021). “Is the propensity to emit alarm calls associated with health status?,” *Curr. Zool.*, **66**, 607–614. doi:10.1093/CZ/ZOAA020
- Noad, M. J., Dunlop, R., Cato, D. H., Stokes, D., Miller, P., and Biassoni, N. (2006). “Humpback whale

- social sounds: Sources levels and response to playback," *J. Acoust. Soc. Am.*, **120**, 3012. doi:10.1121/1.4787051
- Pérez-jorge, S., Tobeña, M., Prieto, R., Vandeperre, F., Calmettes, B., Lehodey, P., and Silva, M. A. (2020). "Environmental drivers of large-scale movements of baleen whales in the mid-North Atlantic Ocean," *Divers. Distrib.*, **26**, 1–16. doi:10.1111/ddi.13038
- Prieto, R., Tobeña, M., and Silva, M. A. (2017). "Habitat preferences of baleen whales in a mid-latitude habitat," *Deep. Res. Part II Top. Stud. Oceanogr.*, **141**, 155–167. doi:10.1016/j.dsr2.2016.07.015
- von Schuckmann, K., Le Traon, P.-Y., Smith, N., Pascual, A., Djavidnia, S., Gattuso, J.-P., Grégoire, M., et al. (2020). "Copernicus Marine Service Ocean State Report, Issue 4," *J. Oper. Oceanogr.*, **13**, S1–S172. doi:10.1080/1755876X.2020.1785097
- Silva, M. A., Prieto, R., Cascão, I., Seabra, M. I., Machete, M., Baumgartner, M. F., and Santos, R. S. (2014). "Spatial and temporal distribution of cetaceans in the mid-Atlantic waters around the Azores," *Mar. Biol. Res.*, **10**, 123–137. doi:10.1080/17451000.2013.793814
- Širović, A., Williams, L. N., Kerosky, S. M., Wiggins, S. M., and Hildebrand, J. a. (2013). "Temporal separation of two fin whale call types across the eastern North Pacific," *Mar. Biol.*, **160**, 47–57. doi:10.1007/s00227-012-2061-z
- Smith, J. N., Goldizen, A. W., Dunlop, R. A., and Noad, M. J. (2008). "Songs of male humpback whales, *Megaptera novaeangliae*, are involved in intersexual interactions," *Anim. Behav.*, **76**, 467–477. doi:10.1016/j.anbehav.2008.02.013
- Stimpert, A. K., Wiley, D. N., Au, W. W. L., Johnson, M. P., and Arsenault, R. (2007). "' Megapclicks ': acoustic click trains and buzzes produced during night-time foraging of humpback whales (*Megaptera novaeangliae*)," *Biol. Lett.*, **3**, 467–470. doi:10.1098/rsbl.2007.0281
- Teixeira, D., Maron, M., and Rensburg, B. J. (2019). "Bioacoustic monitoring of animal vocal behavior for conservation," *Conserv. Sci. Pract.*, **1**, 1–15. doi:10.1111/csp2.72
- Thorley, J. B., and Clutton-Brock, T. (2017). "Kalahari vulture declines, through the eyes of meerkats," *Ostrich*, **88**, 177–181. doi:10.2989/00306525.2016.1257516
- Visser, F., Hartman, K. L., Pierce, G. J., Valavanis, V. D., and Huisman, J. (2011). "Timing of migratory baleen whales at the azores in relation to the north atlantic spring bloom," *Mar. Ecol. Prog. Ser.*, **440**, 267–279. doi:10.3354/meps09349
- Watkins, W. A. (1981). "Activities and underwater sounds of fin whales," *Sci. Rep. Whale Res. Inst.*, **33**, 83–117.

Appendix B

RESPONSE TO REFEREES (ID RSPB-2021-1156)

“Food talk: 40-Hz fin whale calls are associated with prey biomass”

Associate Editor

The authors have done a satisfactory job at addressing the concerns brought up. Incorporating the detection range analysis and adding better descriptions to the methodology have greatly improved the manuscript. In addition, more careful attention to the wording and streamlining the introduction make the study easier to follow. There are, however, a few minor points that are unclear or misleading (see list below with line numbers based on “clean” manuscript). The reviewer also bring up important comments about clarity and flow of the text that deserve attention.

Line 45: If goal is to refer to production of high intensity signals, more clarity is needed.

We thank the editor for this comment. We changed “loudly” by “high intensity” and modified the whole sentence accordingly for clarity.

Line 43: change “cues” to “information”. Cues has a specific meaning in animal communication and it is used to describe stimuli whose perception by other animals is not beneficial to the emitter. Using this word in this sentence is misleading.

We agree with the editor and changed “cues” to “information”.

Line 47-48: what is the second part of the sentence “signal male quality and fitness” adding here? is quality different from body size or health? Also, fitness is unclear and potentially redundant (survival and reproductive ability are different from “quality”?). More clarity is needed here.

We have reworded the sentence and removed fitness as the editor suggested.

Lines 48-57: series of examples about general they provide. Are these sentences just stating that different types of signal provide different information? Without a concluding sentence in this paragraph, it is unclear to the reader what the message is.

We agree with the editor here that these lines did not have a clear message and repeated information from the lines above. Thus, we merged all these lines and now is like “It has been suggested that male songs can convey information about the individual’s reproductive status, body size or health (Nowicki and Searcy, 2004; Tregenza et al., 2006), and may be used by females (Ballentine et al., 2004) and other males to assess the signaller’s quality and competitiveness (De Kort et al., 2009; Moseley et al., 2013)”.

Line 58: while it is it easier to perform manipulations in the systems mentioned compared to working with whales, the wording seems to underplay the amount of work required. I suggest adjusting the wording to better reflect the disparity among different study systems in the use of experimental paradigms that allow researchers to test hypotheses about signal function and content.

We completely agree with the editor about this subject. However, as suggested by the referee, we have deleted this whole paragraph and go straight into talking about fin whale vocalisations.

Lines 69-70: Wording is unclear. is this sentence referring to a comparison to humpback whales?

We acknowledge the sentence was not clear. The paragraph referring to humpback whales has now been deleted and the sentence reworded accordingly.

Lines 285-287: while this is correct, it does not have to necessarily be that way and there are cases in which signals do not provide information about male “quality”. Recognizing that further work that directly test that hypothesis is necessary.

We thank the editor for this comment and have added a few lines and a reference including the suggested information.

Line 301: “fin whale” is twice in this sentence

We thank the reviewer for highlighting this mistake.

Line 316: change “cues” to “information”

We have changed “cues” to “information”

Lines 352-353: remove “future”

“future” has been removed.

Conclusions: the novelty of the study is heavily based on the methodological approach which overshadows the conceptual contribution. While the methods are worth mentioning, I would suggest emphasizing the biological findings rather than the approach.

As the editor suggests, we have emphasized the biological findings by reordering the paragraph and rewording the methodological contribution.

Referee 1

Thanks to the authors for responding to the previous comments and suggestions. The paper is much improved and is much easier to follow as a reader. The paper is more concise and now presents the necessary information. The increased reference to previous literature helps support the work. Below are a few comments/suggestions for edits. Much more of this work is needed and I am encouraged to see the use of acoustics data being presented in this more ecological framework.

Introduction

Lines 42-45: feels disjointed, perhaps rephrase the first few sentences, or starting and line 45 “Males of many species..’ with ‘During reproductive signalling, in many species males...’ instead might help make the link to the previous 2 sentences a little easier. The first paragraph doesn’t quite flow.

We agree with the reviewer that this paragraph did not flow. Thus, we have changed it to: “Animal produce an array of different acoustic signals. These signals can encode various types of information about the signaller’s attributes or external environment, and serve various purposes. During the mating season, males of many species produce high intensity and repetitive songs to attract or court females, repel conspecific males, or both (Amorim et al., 2015; Bennet-Clark, 1971; Payne and McVay, 1971).”

Lines 58-63 in second paragraph are superfluous – you could combine line 63-68 into first paragraph. Or you could just start from Line 69.

As suggested by the reviewer we have deleted this whole paragraph.

Line 79 – why is the ‘40-Hz call’ in quotes here?

We thank the reviewer for highlighting this mistake. We have removed quotes.

Methods

Line 103 – I would think ‘sensor’ needs capitalising

We thank the reviewer for highlighting this mistake.

Line 105 – Try not to start a sentence with an abbreviation

We have changed “EARs” to “Hydrophones”.

Your description of 40-Hz call on Line 109/Line 130 is inconsistent with your description on Line 79

We agree with the reviewer that description for this call was inconsistent. We have now corrected description in the introduction to match that of the methods.

Line 111-112 – as currently written this is a little confusing as to whether it pertains to your data/study, perhaps reword to make it clear the LFDCS has been used previously in other studies/regions etc.

We acknowledge the sentence was confusing and added a reference to our previous study where the LFDCS was used.

Line 120-121 – you say here longer duty cycling recording but the reader has no reference yet that the recordings were duty cycled or how

Duty cycles are now clearly referred to supplementary materials in the first paragraph of methods.

Line 24 – move the reference to the end of the sentence as this refers to blue whale calls

We thank the reviewer for detecting this mistake.

Line 157-138 – resolution of this data? Or is that described line 158-159 – just make sure it is clear

We acknowledge this was confusing and deleted the first mention to resolution and kept the second.

Line 165 – detection range calculation is a result, here a few details of how it was calculated is needed. Reference could made to the supplementary material in the methods, although a paragraph directly stating the calculation would be better. A statement that explains that the call source levels and noise levels used for the calculation were derived the recordings would also be helpful. Some clarification needed on the noise levels used (in Supplementary Materials), were these, for example, the minimum NL for the quietest month and max NL for the nosiest month

We agree with the reviewer that detection ranges are results and should be described in more detail in the methods section. We have now added a section in methods titled “Spatial scale of data integration” that describe calculation of detection ranges. We have also added a section in results titled “Detection range and zooplankton biomass spatial scale”.

Same for Line 167-171 – describe this as a sensitivity test of the scale of data integration, and report the results in the next section

As explained above, we have also added this information in the new section.

You mention the season-prey interaction variable (Line 191) but not any results in the main text. Perhaps just a sentence to state main finding and again refer to Supplementary material.

A sentence has been added in the results section as such: “Although zooplankton biomass varied seasonally (Fig. 2C), the interaction between these two variables had no significant effect on 40-Hz call rates”.

Line 178-193 – be clear with what are results from this study and what are results from previous work that you are now using

Have added some word to clarify if other studies or ours was making the statement.

Line 204-208 – suggest rewording

We have reworded it to make it clearer.

Line 210-211 – lag order of 1 what – this needs a bit more explanation

Have reworded these lines for clarity as such” An autocorrelation at lag one was detected for the 20-Hz call rates, implying there was a correlation between call rates in successive weeks. To account for the temporal autocorrelation, one-week lagged values of 20-Hz call rates were included in the model as a predictor variable”.

Results

Figure 4 – and references to it – make sure it is clear that ‘observations’ is used to indicate call presence

We have replaced “observations” by “observed 40-Hz call rate index”.

Discussion

Line 278: need a period after ‘et al’

We thank the reviewer for detecting this mistake.

Line 282-287 – there are other references that you have made mention too that also support this e.g 43

We have added the reference suggested by the reviewer and also another very recent study (Burkhardt et al., 2021).

Line 288: reads a little awkwardly.

We have reworded it.

Line 292: the difference between 2010 and 2011 is not as striking as you might expect if the result is solely because for whale number/presence

The reviewer may be referring to the difference between 2010 and 2012. We agree with the reviewer this difference is small but still the absence of fin whales in 2012 is a remarkable finding that coincides with lower fin whale singing activity.

Line 296-299 – No data was presented on this here. Maybe present this a future hypothesis to be tested if not going to show data

We have reworded the sentence in accordance to the reviewers comment.

Line 307 – careful, reference 43 does not refer to the backscatter intensity directly as copepod biomass but potential prey

We thank the reviewer for pointing this out and have reword it accordingly.

Line 322 delete ‘foraging competition’ to eliminate some of the repertition

We have deleted foraging competition.

References – need to go through, check formatting etc. (e.g. ref 50)

We thank the reviewer for detecting this mistake. We will carefully revise all references.

REFERENCES

- Amorim, M. C. P., Vasconcelos, R. O., and Fonseca, P. J. (2015). "Fish sounds and mate choice," In F. Ladich (Ed.), *Sound Commun. fishes*, Springer-Verlag, Wien, Vol. 4, pp. 1–33. doi:10.1007/978-3-7091-1846-7
- Ballentine, B., Hyman, J., and Nowicki, S. (2004). "Vocal performance influences female response to male bird song: An experimental test," *Behav. Ecol.*, **15**, 163–168. doi:10.1093/beheco/arg090
- Bennet-Clark, H. C. (1971). "Acoustics of Insect Song," *Nature*, **234**, 255–259. doi:10.1038/234255a0
- Burkhardt, E., Van Opzeeland, I., Cisewski, B., Mattmüller, R., Meister, M., Schall, E., Spiesecke, S., et al. (2021). "Seasonal and diel cycles of fin whale acoustic occurrence near Elephant Island, Antarctica," *R. Soc. Open Sci.*, **8**, 201142. doi:10.1098/rsos.201142
- De Kort, S. R., Eldermire, E. R. B., Cramer, E. R. A., and Vehrencamp, S. L. (2009). "The deterrent effect of bird song in territory defense," *Behav. Ecol.*, **20**, 200–206. doi:10.1093/beheco/arn135
- Moseley, D. L., Lahti, D. C., and Podos, J. (2013). "Responses to song playback vary with the vocal performance of both signal senders and receivers," *Proc. R. Soc. B Biol. Sci.*, , doi: 10.1098/rspb.2013.1401. doi:10.1098/rspb.2013.1401
- Nowicki, S., and Searcy, W. A. (2004). "Song function and the evolution of female preferences: Why birds sing, why brains matter," *Ann. N. Y. Acad. Sci.*, **1016**, 704–723. doi:10.1196/annals.1298.012
- Payne, R. S., and McVay, S. (1971). "Songs of Humpback whales," *Science (80-.)*, **173**, 1441–1447.
- Tregenza, T., Simmons, L. W., Wedell, N., and Zuk, M. (2006). "Female preference for male courtship song and its role as a signal of immune function and condition," *Anim. Behav.*, **72**, 809–818. doi:https://doi.org/10.1016/j.anbehav.2006.01.019